**Gravity Effect of Alpine Slab Segments Based on Geophysical and Petrological Modelling**

Maximilian Lowe [1,2,3], Jörg Ebbing [1], Amr El-Sharkawy [1,4], Thomas Meier [1]

[1] Institute for Geosciences, Kiel University, Germany
[2] NERC British Antarctic Survey, Cambridge, United Kingdom
[3] School of geosciences, University of Edinburgh, United Kingdom
[4] National Research Institute of Astronomy and Geophysics (NRIAG), 11421, Helwan, Cairo, Egypt

Correspondence to Maximilian Lowe: maxwe32@bas.ac.uk

**Abstract**

In this study, we present an estimate of the gravity signal of the slabs beneath the Alpine mountain belt. Estimates of the gravity effect of the subducting slabs are often omitted or simplified in crustal scale models. The related signal is here calculated for alternative slab configurations at near surface height and on a satellite altitude of 225 km.

We apply three different modelling approaches in order to estimate the gravity signal from the subducting slab segments, by: i) Direct conversion of upper mantle seismic velocities to density distribution, which are then forward calculated to obtain the gravity signal. ii) Definition of slab geometries based on seismic crustal thickness and high-resolution upper mantle tomography for two competing slab configurations. The geometries are then forward calculated by assigning a constant density contrast and slab thickness. iii) Accounting for compositional and thermal variations with depth within the predefined slab geometry.

Forward calculations predict a gravity signal of up to 40 mGal for the Alpine slab configuration. Significant differences in the gravity anomaly patterns are visible for different slab geometries in the near surface gravity field. However, different contributing slab segments are not easily separated, especially at satellite altitude. Our results demonstrate that future studies addressing the lithospheric structure of the Alps should have to account for the subducting slabs in order to provide a meaningful representation of the geodynamic complex Alpine area.

Keywords:
Satellite gravity gradient, Alpine subduction, lithospheric and sub lithospheric structure, mantle composition, seismic tomography

**1. Introduction**

Interpretation of gravity anomalies can reveal information on the architecture and tectonic setting of the lithosphere (e.g. Zeyen, & Fernàndez, 1994; McKenzie & Fairhead, 1997; Holzrichter & Ebbing, 2006; Braitenberg, 2015; Spooner et al. 2019). For subduction zones, like the Andes, several studies have shown that the gravity effect of the subducting plates is significant and has to be considered in order to study the feedback between the subducting lithosphere and the overriding plate (Götze et al. 1994; Götze & Krause 2002; Tašárová 2007 Gutknecht et al. 2014; Götze & Pail 2018; Mahatsente 2019). For lithosphere to subduct, a higher density than the surrounding mantle material at the same depth interval is required,

causing a negative buoyancy for the slab and therefore the slab is subducted into earth's interior
(e.g. Kincaid & Olson 1987; Ganguly et al. 2009). However, the gravitational contribution of
subducting material in the upper mantle to the gravity field has so far not been systematically
addressed for the Alpine system. In order to provide an assessment, the magnitude of the gravity
signal of such sub-crustal long wavelength features has to be estimated.
The Alpine mountain belt (Fig. 1a) is chosen for this sensitivity study because firstly a large
range of recent seismic tomography studies imaged subducting slab segments in the Alpine
region (e.g. Babuska et al., 1990; Lippitsch et al., 2003; Spakman & Wortel, 2004; Mitterbauer
et al. 2011; Karousová et al., 2013; Zhao et al., 2016; Kästle et al., 2018; El-Sharkawy et al.,
2020). Those different studies suggest different configurations of slab segments (see section
1.1), allowing us to test how sensitive the gravity field is to varying geometries of subducting
slab segments. Secondly, previous Alpine models addressing the Alpine gravity field have
considered the subcrustal mantle inhomogeneities in form of lithosphere thickness (e.g. Ebbing
et al., 2006; Spooner et al., 2019) or in form of mantle density variations (Tadiello and
Braitenberg 2021), but without identifying the isolated effect of subducting slabs segments in
the velocity or density variations. If the contribution of the mantle density variations is not
considered, a significant part of the gravity field might be attributed to crustal thickness
variations or intra-crustal sources.
In addition, the Bouguer Anomaly of the Alps (Fig. 1b) shows no direct sign of subducting slabs
(in contrast to the Andes subduct zone) as the field is dominated by crustal thickness variations
(Ebbing et al., 2001, 2006). Therefore, forward modeling of the proposed slab geometries, as
imaged by high-resolution tomographic studies, is necesary to separate the gravity signal caused
by the suducting slabs from the gravity anomaly field.

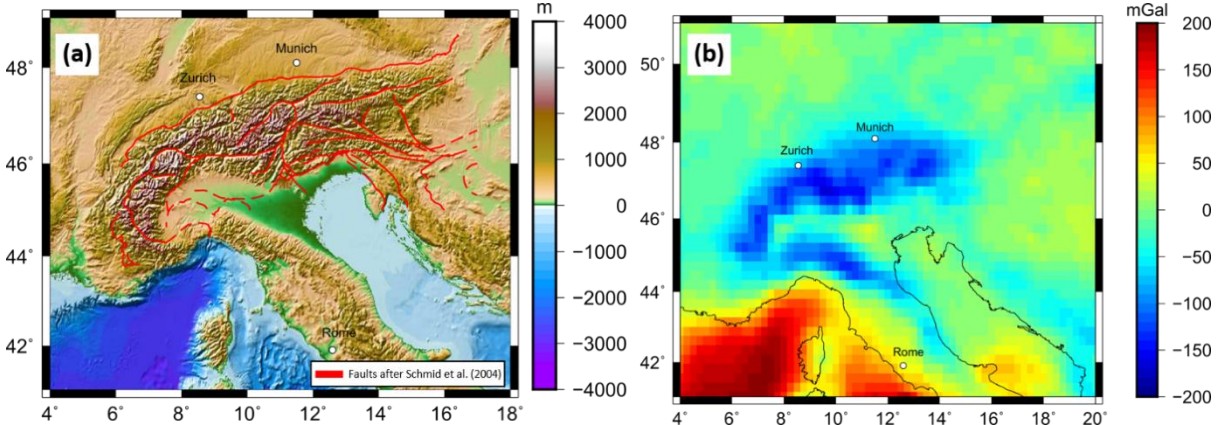

*Figure 1 **(a)** Topography from ETOPO1 from Amante and Eakins (2009). Faults in red after Schmid et al. (2004). **(b)** Bouguer Anomaly based on XGM 2019 (Zingerle et al., 2020) with a maximum spherical harmonics degree of 719 at a station height of 6040m above the ellipsoid, just above the surface of the Alps. Correction density for rock: 2670 kg/m³ and for water: 1030 kg/m³*

We present three different approaches to model the gravity effect of the slab segments and discuss the strengths and limitations of the applied methods. In the first approach, the alpine sup-crustal density distribution is derived by converting seismic velocities to density. This model is then forward calculated to estimate the gravity response. In the second approach, 3D slab geometries are derived by evaluating seismic crustal thickness estimations and high-resolution upper mantle tomographic models. Here, two competing slab configurations are chosen. The predefined slab geometries are then forward calculated by assigning different density contrasts and slab thicknesses. The third approach uses similar predefined slab configurations as in the second approach, however, here we consider petrology, temperature, and density variation. The gravity response is calculated for all three approaches at a near surface height for the gravity disturbance and the gravity gradients at satellite altitude of 225 km.

## 1.1 Alpine setting

The formation and present geodynamics of the Alps are linked to long lasting tectonic processes, including Adria-Europe continent-continent collision, subduction of oceanic and continental lithosphere, the formation of crustal nappes as well as extensional and shortening processes (Frisch, 1979; Stampfli & Borel, 2002; Handy, et al., 2010, 2015). The Adriatic microplate is a major driver of the present geodynamics in the Alpine region, which is trapped

between the converging major plates of Europe and Africa. Adria is moving counterclockwise
with respect to Europe, as seen by GPS observations (e.g. Nocquet and Calais, 2004; Vrabec
and Fodor, 2006; Serpelloni et al., 2016) and is subducted beneath the Apennines to the west as
well as to the east beneath the Dinarides, while colliding with Eurasia in the Alps to the north
(e.g. Channel & Horvath, 1976; Dewey et al., 1989; Stampfli & Borel, 2002; Handy et al., 2010;
Le Breton et al., 2017). Subducting slab segments have been imaged by different seismological
body wave travel time tomographic studies as well as surface wave tomographic studies within
the Alpine upper mantle (e.g. Babuska et al., 1990; Lippitsch et al., 2003; Spakman & Wortel,
2004; Mitterbauer et al. 2011; Karousová et al., 2013; Zhao et al., 2016; Kästle et al., 2018; El-
Sharkawy et al., 2020). However, the configuration of subducting slab segments remains
controversial. In the Western Alps, Lippitsch et al. (2003) propose a slab break-off at about 100
km depth, which is in line with the findings of Beller et al. (2018), Kästle et al. (2018) and El-
Sharkawy et al. (2020). In contrast, a continuous subducting slab segment in the Western Alps,
down to at least 250 km depth, is imaged by a number of other tomographic models (e.g.
Koulakov et al., 2009; Zhao et al., 2016; Hua et al., 2017; Lyu et al., 2017).
A continuous subduction of Eurasia beneath the Central Alps down to at least 200 km depth is
imaged by different tomographic models (e.g. Lippitsch et al., 2003; Piromallo and Morelli,
2003; Koulakov et al., 2009; Mitterbauer et al., 2011; Hua et al., 2017; Fichtner et al., 2018; El-
Sharkawy et al., 2020). A potential slab gap with an approximate size of 2° is separating the
subducting slab segments in the Central Alps to the Eastern Alps as imaged by e.g. Lippitsch
et al. (2003). The slab configuration and subduction direction in the Eastern Alps remains
unclear. According to the classical view, Eurasia is subducting beneath Adria in a southward
subduction (Hawkesworth et al., 1975; Lüschen et al., 2004; 2006). This idea was challenged
by Lippitsch et al. (2003), Schmid et al. (2004), Kissling et al. (2006), Handy et al. (2015), and
Hetenyi et al. (2018). Instead, slab break-off in the eastern Alps and a northward-dipping
Adriatic slab in the easternmost Alps is suggested, leading to a switch of the slab polarity, as

Adria is subducting beneath the European plate (Handy et al., 2015). The view that Adriatic and not Eurasian lithosphere is subducting northwards in the Eastern Alps has been opposed by Mitterbauer et al. (2011), as their model shows a northward dipping slab in the eastern most Alps connected to the European plate. In an early tomographic study, Babuska et al. (1990) proposed that both Eurasian and Adriatic lithosphere is subducting in the eastern Alps. In subsequent studies and interpretations this model was mentioned but northward subduction of Adria seems to be favoured (e.g. Karousová et al., 2013; Hetenyi et al., 2018), Recently, subduction of both Eurasian and Adriatic lithosphere in the eastern Alps down to about 150 km has been suggested by Kästle et al. (2020) and El-Sharkawy et al. (2020) based on surface wave studies. For a more in-depth comparison and discussion of tomographic Alpine models the reader is referred to e.g. Kästle et al. (2020).

**2. Data**

The Bouguer Anomaly (Fig. 1b) is based on the global model *XGM 2019* (Zingerle et al., 2020) developed for spherical harmonics up to degree 719, with a resolution of ~25 km (half wavelength). The XGM 2019 model is a global integrated gravity model, which includes satellite and terrestrial measurements. The Bouguer Anomaly is calculated from the Free-Air gravity disturbance with a correction density of 2670 kg/m$^3$ for topography, and a correction density for water of 1030 kg/m$^3$ for the offshore areas using Tesseroids (Uieda et al., 2016). For the tesseroids, we use the topography and bathymetry from ETOPO (Amante & Eakins, 2009), which was regridded at a regular grid with a grid space of 25 km to match the resolution of the XGM 2019 model for a maximum degree of 719. The gravity field is defined at a constant station height of 6040 m above the ellipsoid, just above the surface of the Alps. The resulting Bouguer Anomaly shows a gravity low in the order of -200 mGal over the high topography of the Alps, indicating an isostatic crustal thickening in response to topography (e.g. Ebbing et al., 2006). Additionally, we calculate the mass correction for the gravity gradients at a station height

of 225 km representing the GOCE satellite altitude. The topographic corrected gravity gradients
after Bouman et al. (2016) measured by the GOCE ESA satellite mission are presented in the
appendix.
For the definition of the slab geometry, we use crustal thickness estimates based on the receiver
function study by Spada et al. (2013). The crustal thickness map was digitized and the Moho
gap in the eastern Alps is filled by nearest neighbour interpolation. To avoid edge effects,
surrounding areas are supplemented by the Moho depth model of the European plate by Grad
et al. (2009), both data sets were merged using a cosine taper with a taper width of 2° using
equation (1). The overlapping areas at the grid edges are distance weighted to obtain a smooth
transition.
$$G_{new} = T(x,y) \cdot G_1(x,y) + \big(1 - T(x,y)\big)G_2 \cdot (x,y)) \quad (1)$$

$$with: T(x,y) = cos\frac{D \cdot \pi}{2 \cdot L}$$

*with: G = Grids, T = taper, D =dx, L = Tapper length*
The merged Moho depth map is sampled at a regular grid with a cell size of 0.25° (Fig. 2) to be
consisted with resolution of the topographic and gravity models.

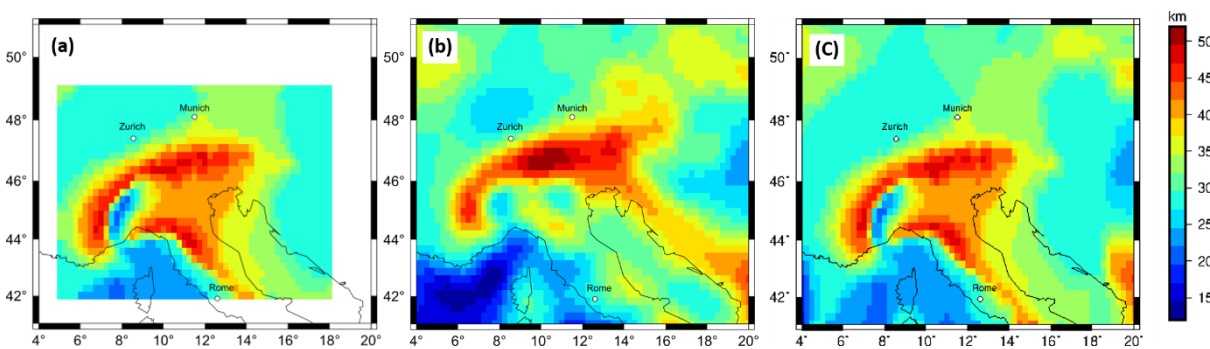

*Figure 2 a) digitized Moho depth after Spada et al. (2013) with a 0.25 ° grid spacing, b) Moho depth estimation after Grad et*
*al. (2009) with a 0.25 ° grid spacing c) Merged Moho depth map from Spada et al. (2016) and Grad et al. (2009) with a grid*
*resolution of 0.25 ° using a cosine taper with a 2° width.*
For the upper mantle seismic velocity, the 3-D shear wave velocity model (MeRE2020) by El-
Sharkawy et al., (2020) is used (Fig. 3). The model covers the upper mantle across the Alpine-
Mediterranean area down to a depth of 300 km and absolute shear-wave velocities are given.
In this study, relative shear-wave velocities in the depth range from 70 to 200 km are calculated
with respect to a 1-D average shear wave velocity model, the background model is described in
El-Sharkawy et al., (2020). The upper limit of 70 km is introduced because i) we focus on the
contribution of the slab segments removing therefore crustal information from the model ii) the
tomography model MeRE2020 is not sensitive to shallow structures, as a result the slabs are
not well recovered in depths shallower than 70 km iii) we want to ensure a uniform upper
boundary. The lower boundary of 200 km is chosen based on clear images of the Alpine Slab
segments to at least 200 km depth (with exception of the Western Alpine slab), as discussed in
section 1, and the assumptions that depth larger than 200 km will have a negligible effect on
the regional gravity field considered here.
The ambient noise tomography by Kästle et al. (2018) is used to define the geometry of the
Western Alpine slab segment, hence we follow the idea of a slab-breakoff in the Western Alps
at 100 km depth (Kästle et al., 2020) as suggested also by Lippitsch et al. (2003) and Beller et
al. (2018). For the eastern Alps, we consider two alternative models. For the first hypothesis,
the P-wave tomography by Lippitsch et al. (2003) is used, to define the Eastern Alpine slab
segment. The second hypothesis is based on Kästle et al. (2020) and El-Sharkawy et al. (2020).
It assumes southward subduction of a short Eurasian Slab as well as northward subduction of a
short Adriatic Slab in the eastern Alps. The slab configurations which are incorporated in the
Alpine density models are discussed in greater detail in section 4.1.
**3 Conversion of seismic velocities into density distribution**
Seismic velocity variations are dependent on temperature and pressure. Densities in the
subsurface are also temperature and pressure dependent. A conversion factor ($\zeta$) can describe
the linear relation between seismic velocities variations and densities variation (e.g. Tiberi et al.
2001; Webb 2009). We convert seismic shear wave velocities from the tomographic model
MeRE2020 by El-Sharawy et al. (2020) in the depth range from 70 to 200 km, as discussed in
section 2, to obtain a density distribution of the upper mantle in the Alpine region based on a
conversion factor ($\zeta$). The relationship between seismic velocities and densities is described in
equation (2), this assumption is a strong simplification of reality, but gives a first order
estimation of the expected relative density structure beneath the Alps.
$$\rho_{rel} = [Vsv_{abs}(1 + \Delta\%) - Vsv_{abs}] \cdot \zeta = Vsv_{abs} \cdot \ \Delta\% \cdot \zeta \quad (2)$$
$with$: $Vsv_{abs} = \ absolute\ velocities\ from\ MeRE2020;$
$\Delta\% = \ percentage\ deviation\ from\ the\ MeRE2020\ background\ model;$
$\zeta = \ conversion\ factor$
The result is strongly dependent on the chosen conversion factor. A range for conversion
factors has been proposed in the literature for different rock types ranging from 0.1 to 0.45 (e.g.
Isaac et al., 1989; Isaak, 1992; Karato, 1993; Kogan and McNutt, 1993; Vacher et al., 1998).
The relative shear-wave velocity distribution in a 3D domain from the tomography model
MeRE2020 from El-Sharkawy et al., (2020) is converted using a constant conversion factor ($\zeta$)
of 0.3. The converted relative density distribution varies between -240 and 350 kg/m$^3$. High
correlations between the structural pattern in the converted density distribution and the relative
seismic velocities are observed (Fig. 3), the similarity in the structure pattern is expected due
to the linear relationship we introduced here. The converted 3D relative density distribution
reflects the variation of seismic velocities in the Alpine lithosphere and therefore includes the
heterogeneities of the subduction slab segments, as seen by the tomographic models (Fig 3).
The relative density model is transferred into tesseroids with a horizontal expansion of 0.2° and
a vertical expansion of 3 km. The Tesseroid model is forward calculated in order to estimate the
gravity response of the converted density distribution of the Alpine lithosphere in the depth
interval of 70 km to 200 km. No horizontal extensions of the mantle model are introduced
because relative densities are used and therefore edge effects are not expected to be significant
and would only affect the outer most degrees of the model. The slab segments are located central
in the model far away from possible artifact due border effects.

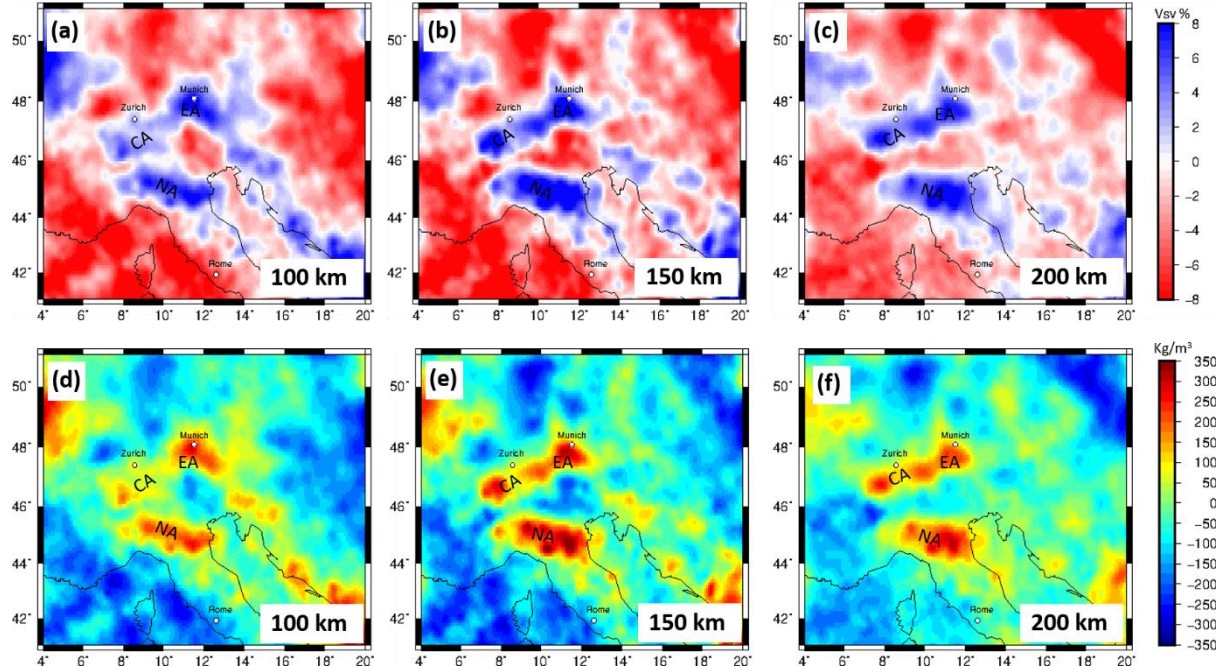

*Figure 3 **(a)-(c)** Depth slices of relative surface wave velocities (Vsv) from MeRE2020 (El-Sharkawy et al., 2020). **(d)-(f)** converted relative density distribution in different depths based on a conversion factor (ζ) of 0.3. CA = Central Alpine Slab; EA = Eastern Alpine Slab; NA = Northern Apennine Slab*

## 3.1 Results

In the forward calculated gravity field, a gravity high with a magnitude of ~40 mGal is observed

over the Alps (Fig. 4). That might be interpreted as relating to the proposed slab segments in

the Northern Apennine and Alpine area. However, the gravity field (and gradients, see appendix)

is dominated by anomalies outside the Alpine realm (Fig. 4), for instance in the Ligurian Sea

and the Dinaride-Hellenide Orogen. Therefore, in the next step, we try to concentrate on the

seismic anomalies in the Alpine realm that can be related to the slab segments.

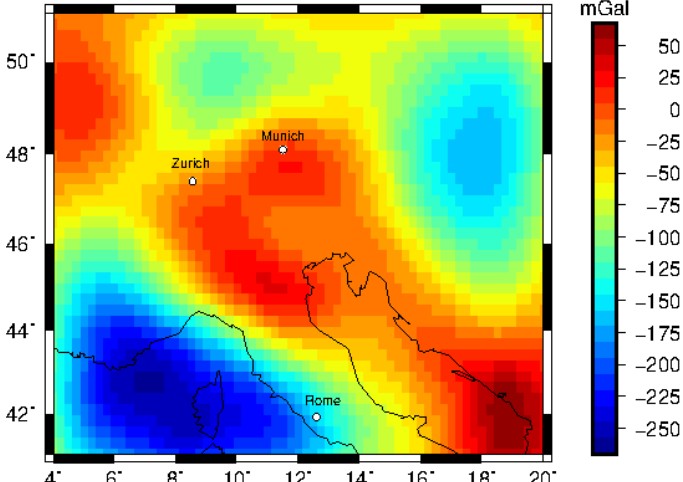


*Figure 4 Forward calculated gravity signal from relative density distribution converted from relative seismic velocities using*
*a conversion factor of 0.3 at a station height of 6040m.*
**4 Slab models**
To estimate the gravity contribution of independent slab segments we introduce different
models for the subducting lithosphere. First, we use a set of models with simple constant density
distribution in the slab, where the parameters, namely the density contrast and thickness of the
slab segment is varied (approach 2). Secondly, we create a set of slab models accounting for
compositional and thermal variations with depth (approach 3). Those models of approach 3 are
created with the software package LitMod 3D (Fullea et al., 2009) and here the slabs are strictly
vertical due to software limitations. Slab models created within LitMod will be referred as
LitMod models in the following. For all non-LitMod models, the gravity and gravity gradients
are calculated using tesseroids, which are spherical prisms (Uieda et al., 2016).
**4.1 Slab modelling with constant density contrast and slab thickness**
We define two alternative slab configurations based on crustal thickness model by Spada et al.
(2013) and several different tomographic studies, see detailed description of the slab
configurations below. At different depths, isolines are picked in the Moho depth map and
tomographic images, defining the upper boundary of subducting slab segments. The isoline of
the crust mantle boundary (Moho interface) is used as an onset of the slab to the crust and
defines the upper boundary of the subduction slab segment. At upper mantle depth, increased
seismic velocity anomalies in tomographic models beneath the Alps are interpreted as contrast
between colder and therefore denser subducting material to the surrounding mantle material. At
100, 150, and 200 km depth, the upper boundary of the slab segment is defined at the 0%
contour line of the relative seismic velocity, marking the transition from rocks with low velocity
to high velocity rocks. The isolines at the Moho interface, 100 km, 150 km and 200 km depth
are displayed upon the Alpine topography (Fig. 5 a-b) Vertical interpolation between the upper
boundary isolines at different depths (Moho depth, 100, 150 and 200 km) define a continuous
surface of the upper slab boundary. The lower boundary of the slabs and therefore the thickness
of the slab segment is not picked based on seismic data but assumed to have constant
thicknesses for simplifications. The thickness is varied for different models from 60 to 100 km
depth.

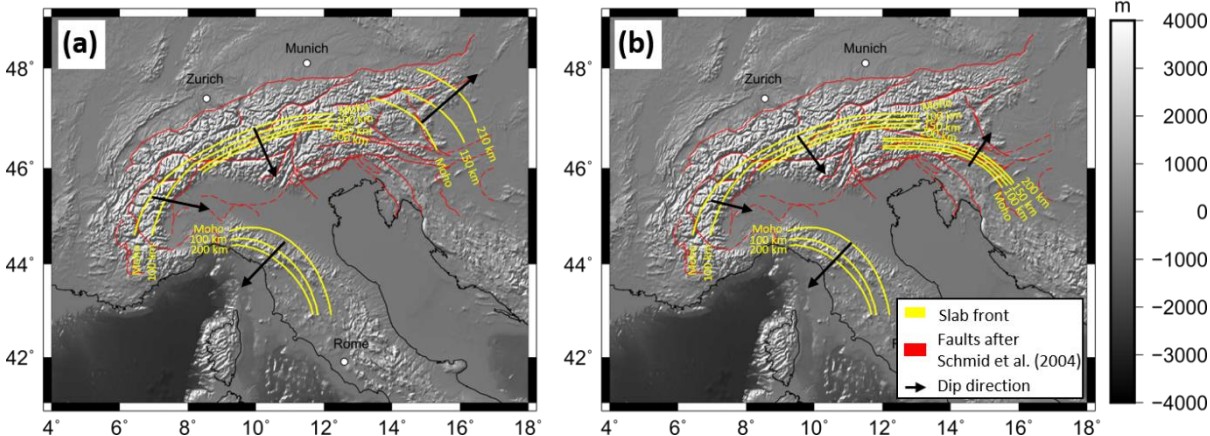

*Figure 5 Defined isolines based on crustal thickness estimations and seismological tomography models for the upper slab*
*boundary for **(a)** Configuration 1 and **(b)** Configuration 2. Black arrows indicate the subduction direction. In red the fault*
*configuration after Schmid et al. (2004).*
3.1.1 Alternative slab configurations
We define two different slab configurations. Configuration 1 (Fig. 5a) features a northeast
subducting slab segment in the Eastern Alps based on Lippitsch et al. (2003). A Central Alpine
slab segment is defined based on Lippitsch et al. (2003) and MeRE2020 (El-Sharkawy et al.,
2020) subducting in south-southeast direction. The Eastern and Central Alpine slab segments
are separated by a slab gap and show perpendicular subduction directions. The east-southeast-
ward subducted slab segment in the Western Alps is defined using the tomographic model of

267 Kästle et al. (2018), supporting the idea of slab break-off at about 100 km depth. Only attached

268 slab segments are considered, ignoring potential mantle upwelling in the break off zone and

269 neglecting the potentially remaining detached slab segment in larger depths. In addition, a

270 southwest-subducting slab segment beneath the northern Apennines is considered down to

271 about 200 km depth, as imaged by MeRE2020 (El-Sharkawy et al., 2020) because of its

272 proximity to the western Alps.

273 Configuration 2 (Fig. 5b) considers a slab configuration mainly based on the interpretation of

274 the MeRE2020 model (Fig. 3) by El-Sharkawy et al. (2020). In the Eastern Alps, both a short

275 southward subducting Eurasian slab segment as well as a short northward subducting Adriatic

276 slab are assumed. The Central and Western Alpine slab segments as well as the slab beneath the

277 northern Apennines are identical to Configuration 1.

278 4.1.2 Forward calculation
279 To estimate the gravity effect of the slab configurations, the geometries are discretized into

280 tesseroids with a 0.2° extension in the horizontal domain and a vertical size of 20 km. The

281 tesseroids range from 40 km to 200 km depth. First, a constant density contrast is assigned to

282 the entire slab. We test density contrasts from 20 kg/m$^3$ to 80 kg/m$^3$. The thickness of the Alpine

283 slab is not well constrained. We test for three slab volumes by assigning three slab thicknesses,

284 60 km, 80 km and 100 km based on studies on other subducting slab segments (e.g. Wang et

285 al., 2020). Due to the curved geometries of the proposed slab segments rectangular tesseroids

286 with a horizontal expansion of 0.2° will either over- or under-estimate the volume of a

287 subducting slab at the edges of the slab. The percentage volume share of each tesseroid to the

288 slab geometry is calculated. The assigned density contrast of the tesseroids which does not lay

289 fully within the slab geometry is decreased according to the percentage volume within the slab

290 geometry. Therefore, the density distribution correlates to the hypothetical slab positions and

291 volumes in the Alpine subsurface without increasing the discretisation resolution of the

292 tesseroid model beyond the uncertainty of gravity measurements and seismic tomographies.

The offset between the 40 km upper tesseroid boundary to the slab onset at the crust in 44 km
depth is corrected using the same process.
4.1.3 Results
Forward calculated slab models for predefined slab geometries of Configuration 1 and 2 with a
constant density contrast of 60 kg/m³ and a constant thickness of 80 km result in a sharp gravity
signal ranging from 70 mGal to 100 mGal (Fig. 6). Both models generate gravity signals in the
order of magnitude of 70 mGal in the Central Alpine region as well as in the Apennine. The
gravity signal in the Eastern Alps differ for the two hypotheses (Fig. 6 a, b). The Western Alpine
slab segment shows the weakest signal in both models.

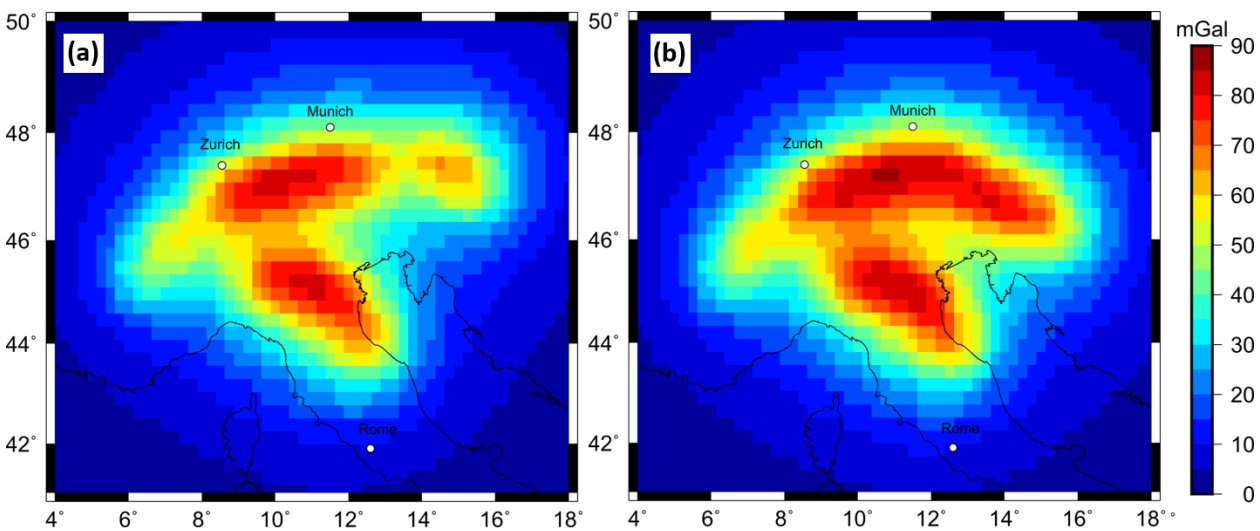


*Figure 6 Forward calculated gravity disturbance signal at a station height of 6040 m for predefined sub-crustal slab geometries*
*with a content density contrast of 60 kg/ m³ and a constant thickness of 80 km. **(a)** predefined slab configuration 1 **(b)** predefined*
*slab configuration 2.*
The gravity signal ranges from 30 to 110 mGal depending on the assigned density contrast and
thickness for both slab geometry models (Fig. 7). The highest magnitude of forward calculated
gravity signal is in the order of 110 mGal and is observed for a slab model with a density
contrast of 80 kg/m³ and a constant slab thickness of 100 km, while the lowest signal is
produced by a combination of 20 km/m³ density contrast and a slab thickness of 60 km. Similar
gravity response is produced by different combinations of density contrast and volume. The
signal pattern is influenced by the predefined slab geometry, while the magnitude of the gravity
signal is depending on the density contrast and thickness (Fig. 7).

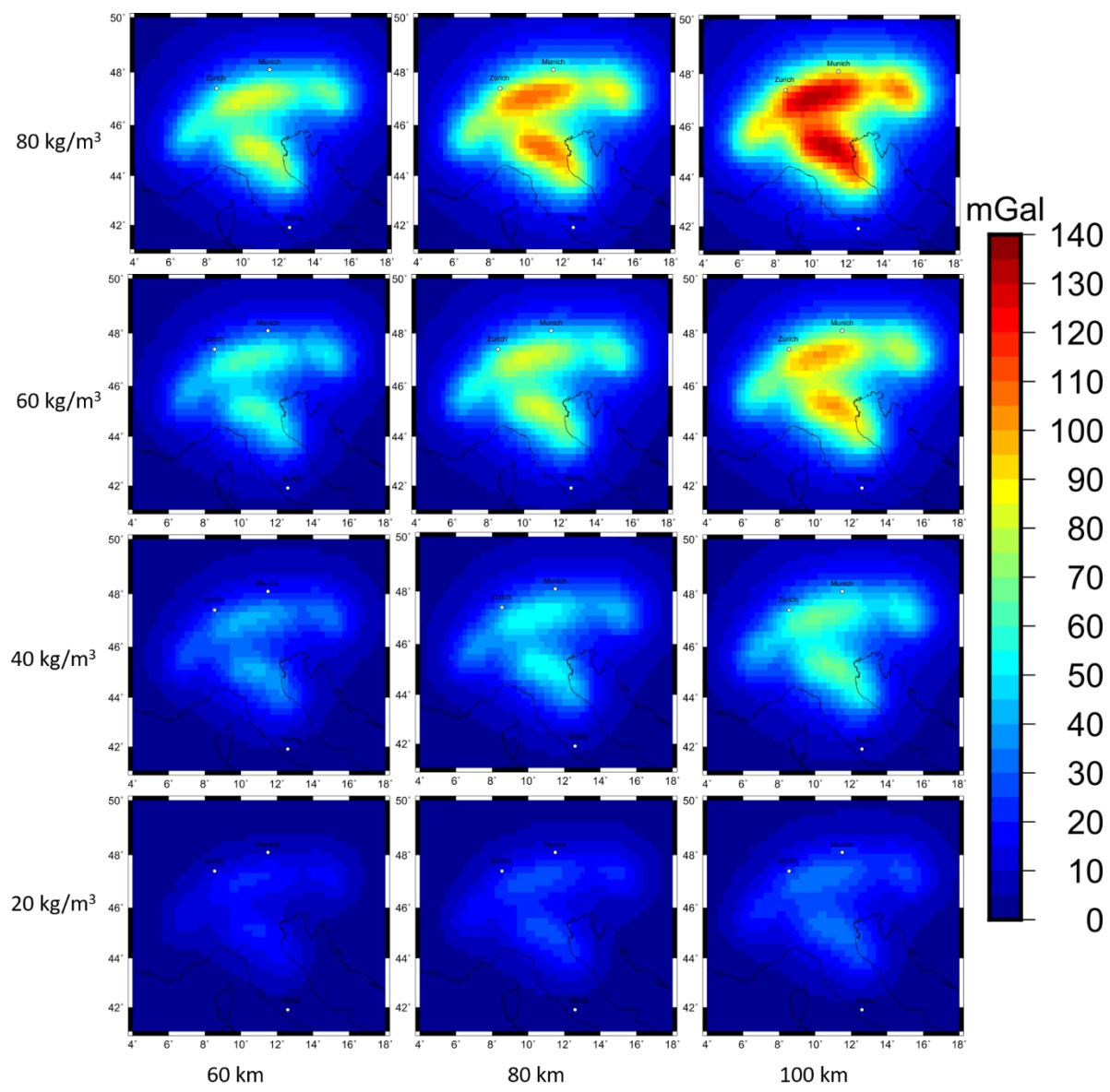


*Figure 7 Forward calculated gravity disturbance signal for 12 different combination of density contrast and slab thickness for*
*sub-crustal slab configuration 1 at a station height of 6040 m.*
Forward calculated gravity gradients at satellite height show the same dependency of signal
strength (see Appendix). The forward calculated gravity field of approach 2 differs significantly
from the forward calculated gravity field of the complete mantle density inhomogeneity of
approach 1 (Fig. 4), which only reaches a positive mantle effect of maximum 50 mGal.
**4.2 Geophysical and petrological modelling with LitMod**
For modelling the Alpine slab segments taking temperature and pressure variations as well as
composition of the lithosphere and sub lithosphere into account, the geophysical and
petrological modelling software LitMod 3D is utilized (Fullea et al.,2009). LitMod 3-D is a
finite difference code, which allows the modelling of lithospheric and sub lithospheric
structures down to 400 km depth by solving the heat transfer, thermodynamical, rheological,
geopotential, and isostasy equations (Afonso et al., 2008; Fullea et al., 2010).
A LitMod model consists of a set of crustal, lithospheric- and sub lithospheric layers
characterized by their petrophysical and thermal properties, which are used as input data (Fullea
et al., 2010). LitMod provides as an output i.e. the density -, temperature-, pressure- distribution
as well as the forward calculated gravity disturbance and gravity gradients (Fullea et al. 2009).
The assigned composition for the different layers is calculated using a LitMod subroutine which
utilizes the Perple_X algorithm of Connolly (2009). Perple_X calculates in the LitMod
implementation the specific bulk rock properties based on the six main lithospheric oxides
($SiO_2$, $Al_2O_3$, FeO, CaO, $Na_2O$) by minimizing Gibbs free energy equation. The Alpine
lithosphere and sub lithosphere as well as the proposed slab segments are modelled using
standard global lithospheric and sub lithospheric compositions to test the influence of
compositional variations within the slab segments on the gravitational signal. Here, we use the
so-called Tecton and Proterozoic type-composition (Table 1). Those compositions were chosen
for a model with a homogeneous crust, lithosphere and sub-lithosphere, where the density
changes as a function of temperature and pressure based on the assigned compositions. The
different slab composition is introduced to test whether a compositional contrast, in addition to
the expected thermal difference, results in a significant density contrast between the slab and
the surrounding material.
*Table 1: Mineralogical composition for the lithospheric and sub lithospheric structure.*

| Major Oxide Compositions | Aver. Tecton Gnt. SCLM [a] | Aver. Tecton Gnt. Peridotite [a] | Average Proterozoic Massif | PUM [b] | DMM [c] |
|---|---|---|---|---|---|
| $SiO_2$ | 44.5 | 45 | 45.2 | 45 | 44.7 |
| $Al_2O_3$ | 3.5 | 3.9 | 2 | 4.5 | 3.98 |
| FeO | 8 | 8.1 | 7.9 | 8.1 | 8.1 |

| MgO | 39.8 | 38.7 | 41.6 | 37.8 | 37.8 |
|---|---|---|---|---|---|
| CaO | 3.1 | 3.2 | 1.9 | 3.6 | 3.17 |
| Na$_2$O | 0.26 | 0.24 | 0.13 | 0.36 | 0.13 |

[a] Classifications according to Griffin et al. (1999b), [b] McDonough & Sun (1995), [c] Workman & Hart (2005) DMM = Depleted
mid-oceanic ridge basalt mantle, PUM = primitive upper mantle.
First, we create a reference model (M$_0$) without a slab segment. This model contains topography
from the ETOPO1 dataset (Amante and Eakins, 2009), the Moho depth from Spada et al. (2013)
and Grad et al. (2009). The lithosphere asthenosphere boundary (LAB) is a required interface
for the LitMod 3D to divide the model between the lithosphere and sub lithosphere and to assign
compositions. We introduce a fixed technical LAB at a depth of 100 km throughout the model
despite of the presence of slabs as the LAB is defined as the 1300°C isotherm. This set-up avoids
that the isotherm follows the geometrical shape of the slab, which would lead to a location in
unrealistic large depths (>200 km). In addition, we neglect the topography of the LAB for
several reasons: i) the information of the lithospheric thickness in the Alpine forelands is spare
and under ongoing discussions, ii) the fixed depth value is based on thermal isostasy LAB
estimations from Artemieva et al. (2019), which shows a LAB depth in the range of 80 to 120
km depth in the Alpine forelands. This technical LAB is used to parameterize the model and is
not meant to represent the topography of the LAB. The modelled slab segments are extending
vertically downwards.
Slab segments are introduced stepwise for the lithosphere and sub lithosphere domains into the
model as well as thermal anomalies for the slab segment beneath the technical LAB, which
describes the 1300°C isotherm (Table 2). Calculating the difference to the reference model (M$_0$)
allows to estimate the effect a slab segments has on the density, temperature distribution of the
Alpine subsurface and therefore on the Alpine gravity field based on slab position, slab
geometry and composition.
*Table 2: Different LitMod models and there incorporated lithospheric and sub lithospheric structures and compositions.*

| Models | Slab geometries | Slab composition (mantle) | Mantle composition | Slab composition (sub lithosphere) | Sub lithosphere composition | Thermal anomaly within sub lithospheric slab |
|---|---|---|---|---|---|---|
| $M_0$ | - | - | aver. Tecton Gnt. | - | PUM | - |
| $M_1$ | Configuration 1 | Aver. Tecton Gnt. Peridotite | aver. Tecton Gnt. | - | PUM | - |
| $M_2$ | Configuration 2 | Aver. Tecton Gnt. Peridotite | aver. Tecton Gnt. | - | PUM | - |
| $M_3$ | Configuration 1 | Aver. Tecton Gnt. Peridotite | aver. Tecton Gnt. | DMM | PUM | -100 °K |
| $M_4$ | Configuration 2 | Aver. Tecton Gnt. Peridotite | aver. Tecton Gnt. | DMM | PUM | -100 °K |
| $M_5$ | Configuration 1 | Aver. Tecton Gnt. Peridotite | aver. Tecton Gnt. | PUM | PUM | - |
| $M_6$ | Configuration 1 | Aver. Tecton Gnt. Peridotite | aver. Tecton Gnt. | DMM | PUM | - |
| $M_7$ | Configuration 1 | Aver. Tecton Gnt. Peridotite | aver. Tecton Gnt. | DMM | PUM | -200 °K |
| $M_8$ | Configuration 2 | Average Proterozoic Massif | aver. Tecton Gnt. | - | PUM | - |

A positive density contrast between subducting material and the surrounding mantle material
results in a negative buoyancy force. A density contrast is introduced into the LitMod model by
a difference in composition between the subducting denser slab and the surrounding mantle
(Fig. 9). Here, we use Tecton like compositions for the lithosphere and the subducting slab
segments since the Alpine slab segments result from continent-continent collision (Tables 1 and
2). A later model features a Proterozoic slab composition ($M_8$). Depleted mid-oceanic ridge
basalt mantle (DMM) and primitive upper mantle (PUM) are used for the sub lithospheric
domain. Additional to the density contrast within the sub lithosphere, a temperature anomaly of
– K is introduced for the sub lithospheric part. Later models include a variation of
temperature anomalies ($M_5, M_6, M_7$). Note those compositions are used as a first order test and
serve as a starting point for synthetic slab models to illustrate the compositional and thermal
effect on the gravity signal by influencing the density distribution. They do not necessary
represent the compositional mantle environment in the Alpine region.

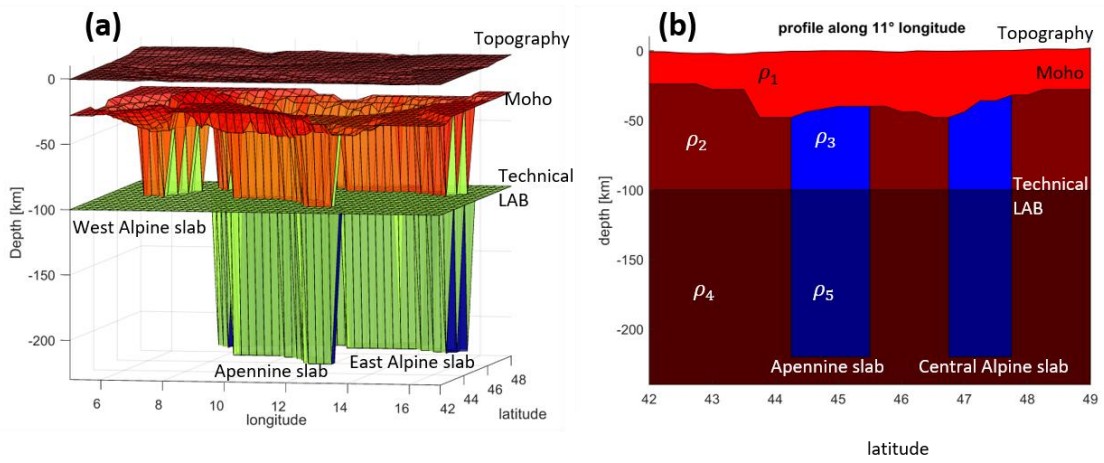


*Figure 8 **(a)** 3D model set up using LitMod 3D. Topography, Moho and LAB depth as well as the vertical incorporated slab models are used as input layers with assigned petrophysical and thermal properties. **(b)** Profile along 11° longitude through a LitMod model containing Topography, crustal and lithospheric thickness as well as a slab segment. $rho_{1-5}$ indicate petrophysical and thermal property variations for each layer.*

4.2.1 Results

The gravity signal of the predefined slab segments is forward calculated as well as the

background model without incorporation of slab segments. The residual between both forward

calculations gives the gravitational contribution of the slab segments, while other gravitational

effects, like the topography or crustal thickness variation and mantle variations outside the slab

are not considered.

A slab segment with an average Tecton Gnt. composition ($M_1$, $M_2$) results in a slightly denser

material compared to the surrounding mantle ($M_0$), while a slab segment with a Proterozoic

composition ($M_8$) shows a less dense lithospheric structure compared to the reference model

($M_0$), this composition results in less dense slab segment, which would not be subducted due to

the positive buoyancy (Fig. 10). However, we aim to illustrate the effect composition has on

the density distribution within the slab and to the surrounding mantle and show the importance

of correct compositional information, therefore we focus on the difference in density contrast

between slab and surrounding mantle and neglecting the sign of the density contrast.

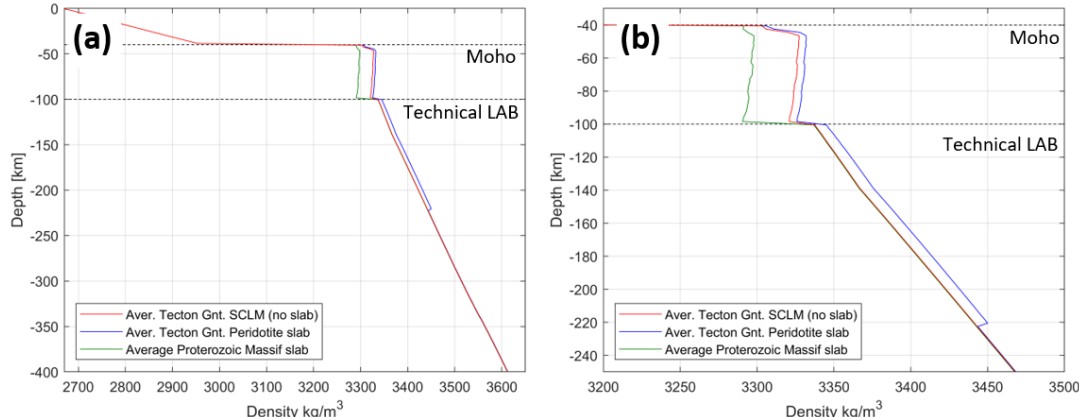

*Figure 9 (a) density profile at 11° longitude and 45° latitude for the full vertical model space of 400 km depth. Density profiles for 3 different models (M0, M1, M9) with different compositional properties are shown. (b) Zoomed in profile at the depth range of present slab segments.*

The difference in density distribution (density contrast) within the slab segments with a Tecton composition ($M_1, M_3$) to the reference model ($M_0$) is in the order of 5 kg/m$^3$ for the lithosphere and in the order of 10 kg/m$^3$ for the sub lithospheric domain (Fig. 10a). The density variations within the lithospheric and sub lithospheric slab domain are less than 1 kg/m$^3$ resulting from both depth dependent variations in pressure and temperature. Between lithosphere and sub lithosphere, a rapid increase in density contrast is observed (Fig. 10a). The density contrast of a lithospheric Proterozoic slab composition ($M_9$) to the reference model ($M_0$) is in the order of -30 kg/m$^3$ (Fig. 10b).

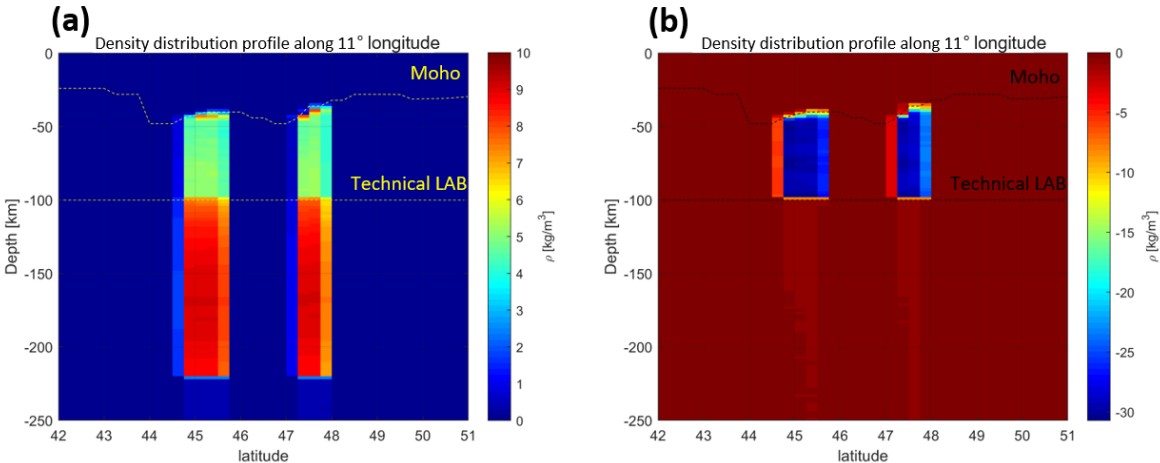

*Figure 10 (a) residual density contrast for lithospheric and sub lithospheric slab segments of model (M3) with Tecton like composition within the lithosphere and PUM and DMM composition in the sub lithosphere with an additional thermal anomaly of -100° k for the sub lithospheric slab segment to the background model (M0). (b) Residual lithospheric density contrast of a Proterozoic lithospheric slab segment (M8) to a Tecton compositional surrounding mantle (M0). Residual density contrast is limited to the technical LAB as the sub-lithospheric part is identical to the reference model (see also Fig. 9b)*

The gravity signal caused by the proposed slab segment configurations is estimated for
lithosphere and sub lithosphere separately. The forward calculated gravity effect, at topographic
surface level, for the slab configuration 1 for the lithospheric part is in the order of 4 mGal
while the sub lithospheric gravity signal is in the range of 7 mGal (Fig. 12a, b). The combined
gravity signal is in the order of 12 mGal (Fig. 12c). The gravity signal in the Eastern Alps for
Configuration 2 is significantly larger in the order of 17 mGal for the combined model (Fig.
12f).

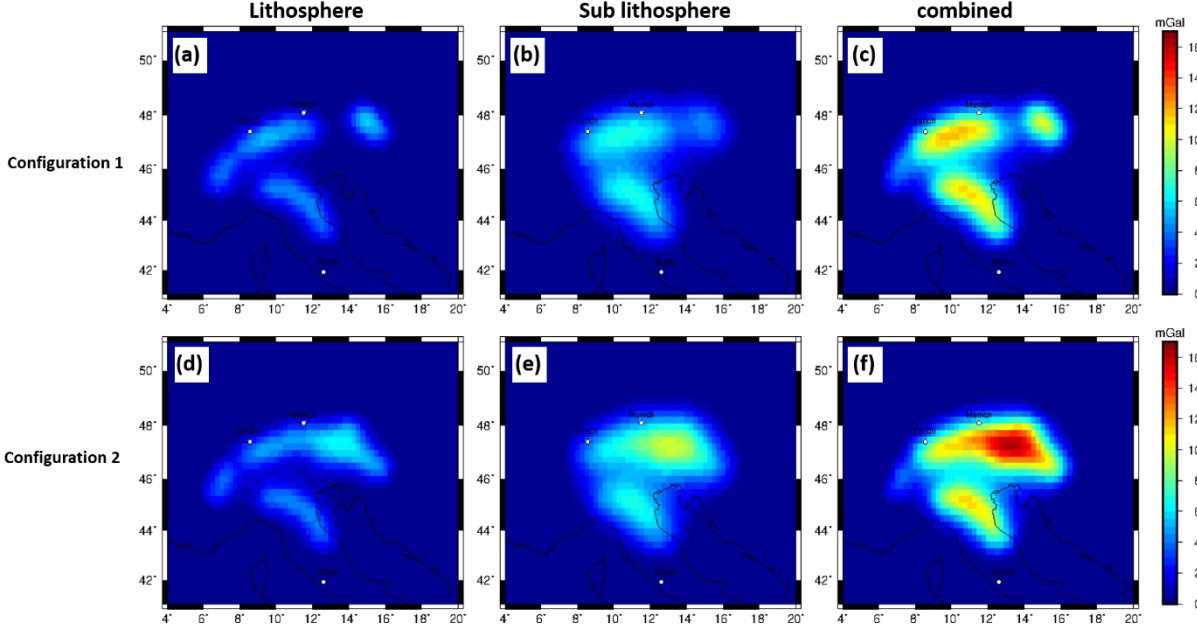


*Figure 11 Residual of the forward calculated gz gravity signal of lithospheric slabs at surface station height based on LitMod*
*models with Tecton like compositions in the lithosphere and PUM and DMM compositions in the sub lithosphere (M1, M2, M3,*
*M4,) with an additional thermal anomaly of -100° k for the sub lithospheric slab segment, for predefined slab Configuration to*
*the background model (M0). **(a)-(c)** Configuration 1. **(d)-(f)** Configuration 2. Crustal and topographic contribution are nullified.*
The calculated gravitational effect of a slab segment with Proterozoic composition and a Tecton
surrounding mantle composition is in the order of -40 mGal for the gz component (Fig. 12 a).

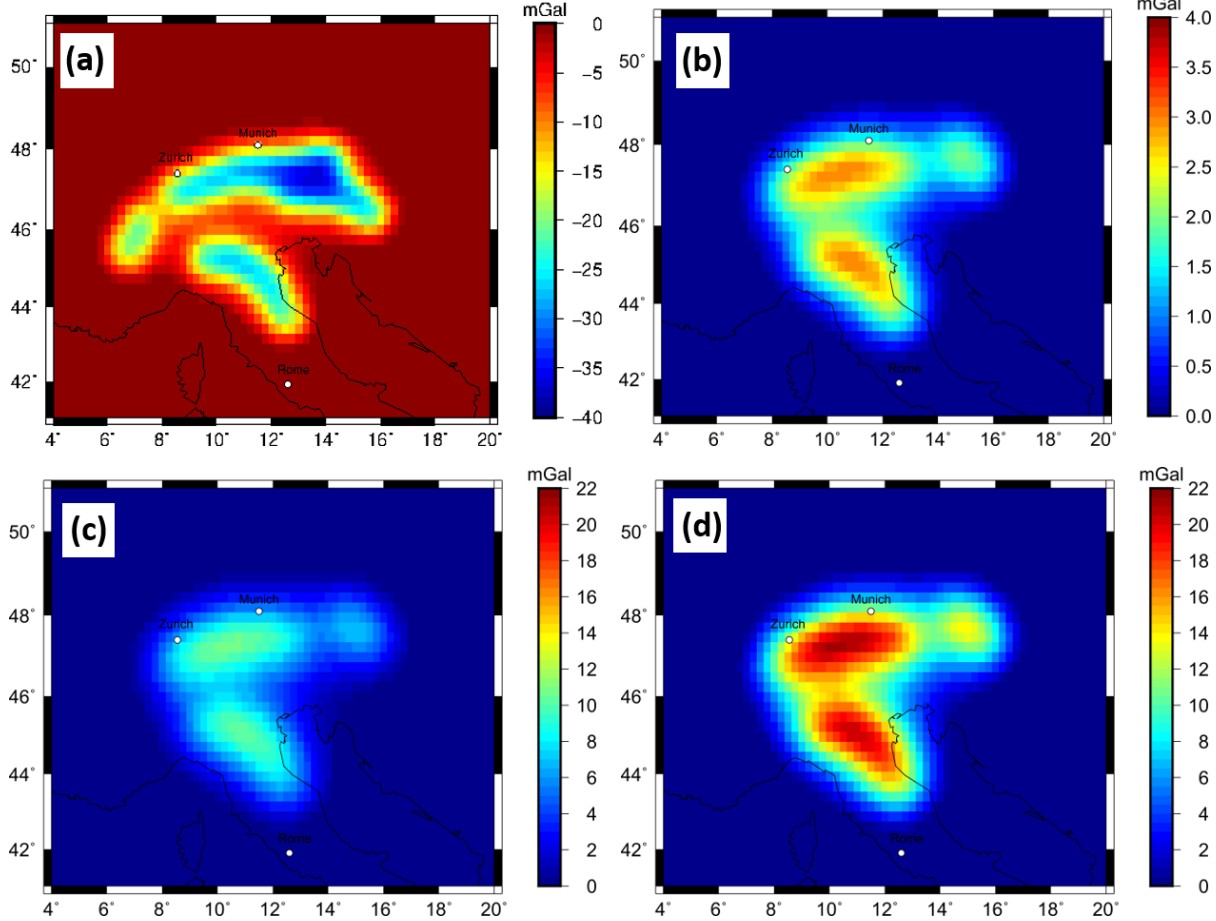

*Figure 12 (a) Forward calculated gravity effect of a Proterozoic lithospheric slab segment to a Tecton compositional*
*surrounding mantle for configuration 2, obtained by calculating the residual between $M_8$ and $M_0$. (b) gravity signal produced*
*by purely compositional effect in the sub lithosphere between a PUM and DMM composition, obtained by calculating the*
*residual between $M_5$ and $M_6$. (c) gravity signal produced by purely thermal anomaly of -100° K for a sub lithospheric slab*
*segment, obtain by calculating the residual between $M_3$ and $M_6$. (d) gravity signal produced by purely thermal anomaly of -*
*200° K for a sub lithospheric slab segment obtained by calculating the residual between $M_6$ and $M_7$.*
The gravity response to a compositional variation within the sub lithosphere between the
incorporated slab segment (DMM composition) and the surrounding mantle (PUM composition)
is in the order of 4 mGal (Fig. 12b). The gravity response for a pure thermal anomaly of -100
K within the sub lithospheric slab segment is in the order of 16 mGal (Fig. 12c), while a pure
thermal anomaly of -200 K within the sub lithospheric slab segment is in the order of 21 mGal.
**5 Discussion**
The imprint of the gravity response caused by the density distribution based on direct
conversion of seismic velocities (approach 1) is visible, however, individual and independent
slab segments cannot be identified (Fig. 4). The strength of this approach is that it is fast to

implement and can provide a first order characterization of the gravity signal and slab geometries of subducting lithosphere. However, a clear characterization of subducting slab segments is not possible. First of all, the density model depends on the resolution and regularization of the seismological model, which can lead to distortions in the gravity response (e.g. Root, 2020). The method is dependent on the choice of the conversion factor and might overestimate the density (see the large negative anomaly in the Ligurian Sea). The conversion factor is a strong simplification of nature and for such a geodynamic complex area, a constant conversion factor is not adequate.

The forward calculated gravity field with competing predefined slab geometries (approach 2) shows a clear gravity signal, where the individual slab segments are distinguishable (Fig. 6). A relative gravity low related to the slab gap in the Eastern Alps is a prominent feature in the gravity signal of Configuration 1 (Fig. 6a). The eastern Alpine slab segment of Configuration 1, due to its relatively small volume, result in a lower signal compared to the Central Alpine slab segment.

Configuration 2 shows a larger gravity signal in the Eastern Alps up to 100 mGal (Fig. 6b) compared to Configuration 1. The increase of the gravity signal is attributed to the subduction of both Eurasian and Adriatic lithosphere in the Eastern Alps. The gravity signal shows a continuous transition from the Central Alps to the Eastern Alps, where the contribution of the destined slab segment cannot be distinguished in the resulting gravity field (Fig. 6b). In the Western Alps, Configuration 1 and 2 show a lower gravity signal compared to the Central Alps. This is attributed to the much shallower Western Alpine slab segment that penetrate down to 100 km depth.

The gravity signal is influenced by both the assigned density contrast and thickness of the slab. A trade of between both parameters is clearly observable as the same gravity response of the slab configuration can be achieved with different values of density contrast and slab thickness.

Therefore, making it impossible to derive slab properties in form of density contrast and slab
thickness from the gravity field (Fig. 7).
The calculated densities in LitMod 3-D models (approach 3) are estimated taking temperature
and pressure variations into account based on an assigned composition. The composition has a
strong influence on the resulting density contrast. In the case that the compositional contrast
between slab segment and surrounding mantle is small, the density contrast is consequently
small as well (Fig. 9 and 10a). With increasing compositional differences, the density contrast
increases as well. A strong density contrast within the slab segment is recognizable between
lithospheric and sub lithospheric domain (Fig. 10a and b), while the variations between the slab
and surrounding mantle remain small.
The gravity signal shows in the Eastern Alps significant larger signal from the lithosphere and
sub lithosphere domain for Configuration 2 (Fig. 11d, e, and f) compared to Configuration 1
(Fig. 11a- c). The different slab segments are distinguishable with the exception of the two slab
segments in the Eastern Alps in Configuration 2 (Fig. 11). The contribution from the
lithospheric domain to the gravity signal is smaller than from the sub lithospheric domain (Fig.
11b, and e). However, the slab gap and the eastern slab segment feature can be recognized in
the lithospheric part in Configuration 1 but not in the gravity signal of the full model.
The Proterozoic slab segment has a larger gravity response compared to the Tecton-like
composition. This gravitational signal is negative due to the less dense Proterozoic composition
in comparison to the reference model ($M_0$) (Fig. 12a).
Sub lithospheric composition has only a small influence on the gravity field, in the order of 4
mGal (Fig. 12b). However, a thermal anomaly within the sub lithospheric slab in the order of -
100K result in a gravitational response of 16 mGal (Fig. 12c) and for a -200 K anomaly in the
order of 21 mGal (Fig. 12d). Both the composition and the thermal variation influence the
density and consequently the gravity response. However, the thermal component is a much
larger contributor.
For the three approaches (section 3, 4.1 and 4.2) a measurable gravity effect of the subducting
slab segments is observable. The independent slab segments are distinguishable to a certain
degree with the exception of the bivergent slab configuration in the Eastern Alps (Fig. 6, and
11) and the model containing converted density from seismic velocities (Fig 4), while the slab
configurations cannot be separated at satellite altitude (Appendix). Forward calculated gravity
anomalies from converted density distribution suggest a gravitational signal of the slab
segments in the order of 40 mGal which corresponds to a density contrast of 20 to 40 $kg/m^3$ in
the models with predefined slab geometry. The models with a Tecton like composition suggest
a gravity effect of the slab segments in the order of only 16 mGal, corresponding to a density
contrast of 20 $kg/m^3$ in the simple model. Increasing the compositional difference with a Tecton
composition suggests a gravity signal in order of 30 mGal and is in line with the converted
density model.
All three methods show a positive gravity signal contribution, which can be related to sub-
crustal density variations for approach 1 and to predefined sub-crustal slab segments for
approach 2 & 3, up to 40 mGal to the Alpine gravity field. That is significant in comparison to
the observed Bouguer Anomaly with a minimum of ~-200 mGal. If this contribution is not
considered, a significant part of the gravity signal is attributed to crustal thickness or intra-
crustal sources. Due to the long-wavelength appearance of the gravity effect, that might not be
relevant for small-scale or local studies, the effect is only seen as a shift. For gravity models of
larger areas (e.g. Eastern Alps) or even the entire regions that should not be neglected. For one,
estimates of crustal thickness or the mass distribution are significantly biased, and placing the
Alps in the geodynamic context of the surrounding requires a careful and complete
consideration of all sources in order to provide realistic density distribution required for
geodynamic models (e.g. Reuber et al., 2019).
**6 Conclusions**
We have addressed the potential gravity effect of proposed slab segments in the Alpine region
using three different modelling approaches.

• Converted density from seismic tomography: In the resulting gravity signal the imprint

of slab segments is visible, however, distinguishing between the different and

independent slab segments is not possible.

• Models with predefined slab segments are dependent on the assigned density contrast

and volume as well as on the predefined positions of the slab segments. The gravity

signal caused by the slab segments are sharp and can be separated for the different slab

segments for the gravity field at the surface. Significant gravity contributions from slab

segments below 200 – 250 km to the Alpine gravity are unlikely.

• Combined petrophysical-geophysical modelling results in the most complex models.

The calculated density variation within the slab is rather small compared to the density

contrast between lithosphere and sub lithosphere. The density distribution within the

slabs, and consequently the gravity field, is highly influenced by the slab composition

and thermal structure.

Sub-crustal density variation (approach 1) and predefined slab segments (approach 2 & 3)
suggest a positive sub-crustal gravity contribution of up to 40 mGal. Even though this might be
considered as a maximum gravity estimation of slabs, this value is significant, even compared
to the observed Bouguer Anomaly low of -200 mGal along the Alps. The interpretation of
density variation in the mantle in terms of subducting slab structures is a means to provide a
meaningful representation of the geodynamic complex Alpine area. For future studies correct
slab density structure are crucial to provide a representation of the Alpine geodynamic setting.
Precise estimations of the slab density structure require a correct crustal density and crustal
thickness model. With the integration of further observables, it might be possible to judge on
the correct slab configuration beneath the Alps. Furthermore, future studies based on the
AlpArray Network will be of high interest in better defining slab geometries as well as their
properties.
**Competing interests**
The authors declare that they have no conflict of interest.
**Author contributions**
ML carried out the gravity modelling, visualized and interpreted the results and prepared the
first manuscript draft. JE supervised the gravity modelling and interpretation, designed the
original research project, acquisition of the financial support for the project leading to this
publication and writing (reviewing and editing). TM defined the slab configurations based on
tectonic and seismological knowledge and writing (reviewing and editing). AE created and
provided the surface wave tomography model MeRE2020 and writing (reviewing and editing).
**Acknowledgment**
The authors thank the reviewers, Carla Braitenberg and an anonymous referee for their valuable
suggestions, which helped to improve the manuscript significantly.
This study is part of the projects "Integrierte 3D Modellierung des Schwere- und
Temperaturfelds zum Verständnis von Rheologie und Deformation der Alpen und ihrer
Vorlandbecken - INTEGRATE" and "Surface Wavefield Tomography of the Alpine Region to
Constrain Slab Geometries, Lithospheric Deformation and Asthenospheric Flow in the Alpine
Region" funded by German Research Foundation (DFG) in the SPP Mountain Building
Processes in 4D.
We thank the developer of open scientific Software which were utilized in this study: tesseroids
(Uieda et al., 2016), LitMod 3D (Fullea et al., 2009 and Afonso et al., 2008) and Generic
Mapping Tools (GMT) (Wessel et al., 2013; Wessel & Luis, 2017).

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

**Appendix A Gravity Gradients at satellite height**
For all Alpine density models presented above (section 3, 4,1 and 4.2) we have also calculated
gravity gradients at a station height of 225 km. This station height corresponds to the second
mission phase of GOCE (Gravity field and steady-state ocean circulation explorer) carried out
by ESA (European Space Agency).
We anticipated that gravity gradients measured by the GOCE satellite mission are sensitive to
the slab segments in the Alpine region. Our result show, that the long wavelength signal of the
different present slab segments contributes to a large-scale gravity response where the different
contributor cannot be separated. Therefore, we conclude that against our anticipation gravity
gradients at satellite height are in fact not sensitive to the Alpine slab configuration. We show
here, the gravity gradients (mainly the gzz component) for completeness.
Measured gravity gradients from the GOCE mission (Bouman et al., 2016), which were
corrected for topography and bathymetry ranges from 2.5 to -2.5 E at satellite altitude of 225
km height (Fig. 13). A negative gravity anomaly of -2.5 E in the gzz component is observed
equivalent to the vertical gz component (Fig. 13). However, no clear sign for subducting
lithosphere can be observed in any component of the gravity gradient tensor.

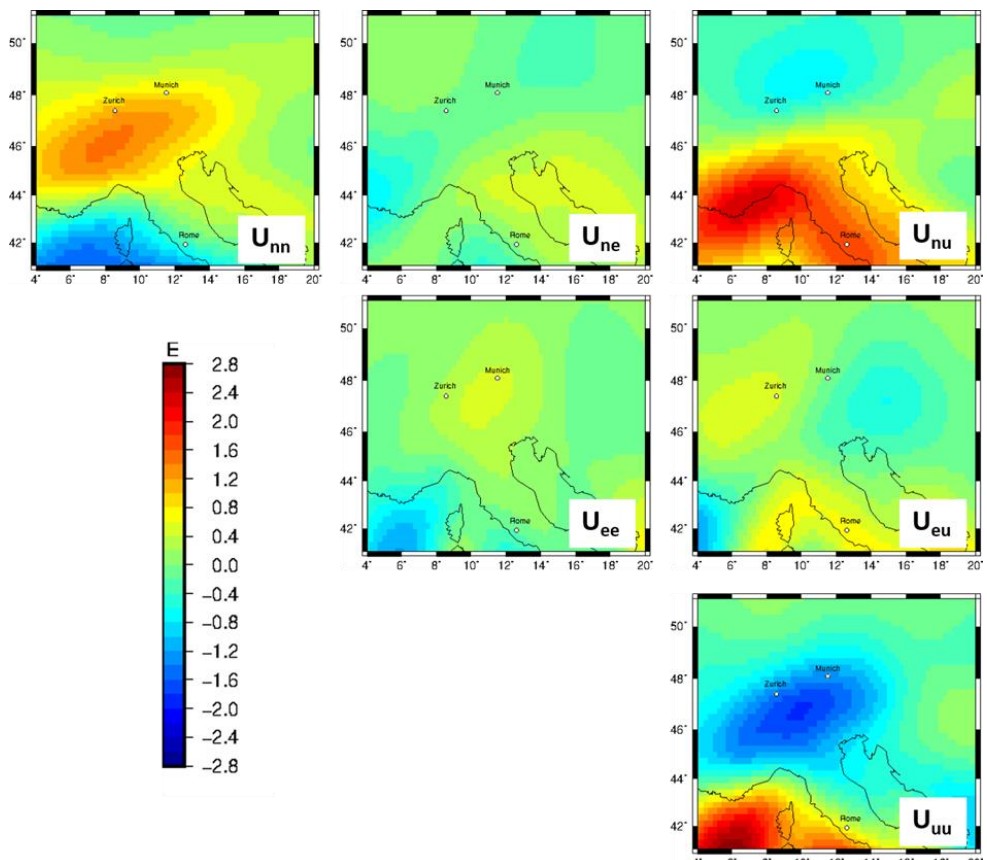

*Figure 13 GOCE gradients at 225 km height after Bouman et al. (2016) corrected for topography and bathymetry with a 5°*
*extension to remove far filed effects. The gravity gradients are presented in a North-East-Up coordinate system.*
The forward calculated gzz component at 225 km station height from a density model (section
3) with converted densities ranges from -3.5 E to 0.7 E (Fig. 14). A positive gravity signal of
about 0.5 E in the Apennine and Alpine region is observed which could be linked to subducting
slab segments. However, it is impossible to separate specific slab segments.

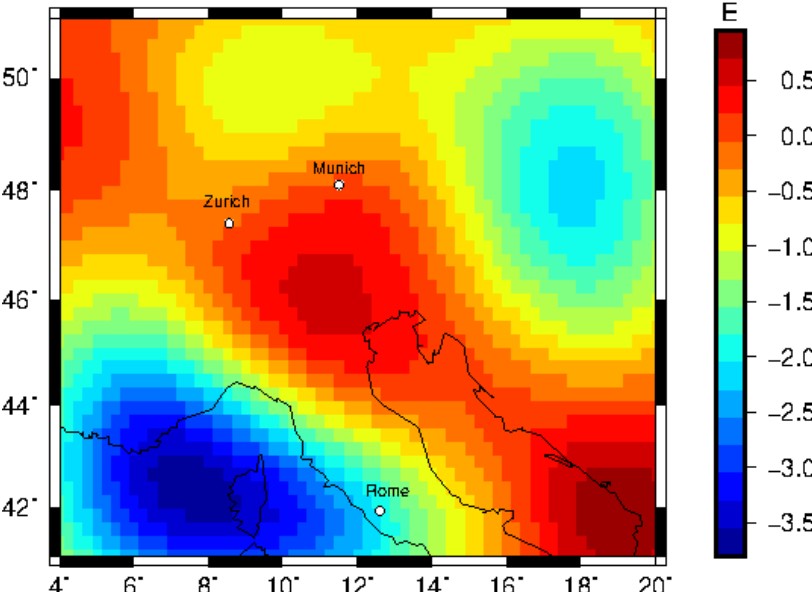

*Figure 14 Forward calculated gzz gravity signal from relative density distribution converted from relative seismic velocities*
*using a conversion factor of 0.3.for the at 225 km station height.*
Forward calculated tesseroid models (section 4.1) for slab configuration 1 and 2 with a constant
density contrast of 60 kg/m$^3$ and a constant thickness of 80 km result in a less sharp gravity
signal for the gzz component at a station height of 225 km (Fig. 15) compared to the gz
component at station height of 6040 m (Fig. 6). The gravity signal for the gzz component is in
the range of 0.8 E to 1 E. At satellite altitude the gravity signal is observed as a large area with
a positive gravity effect for Configuration 1 and 2. The contribution of the different slab
segments to this positive gravity effect is not distinguishable. The only recognizable difference
is the size of this positive gravity signal. Configuration 1 shows a smaller anomaly, due to a
lower volume of subducting material in the Eastern Alps.

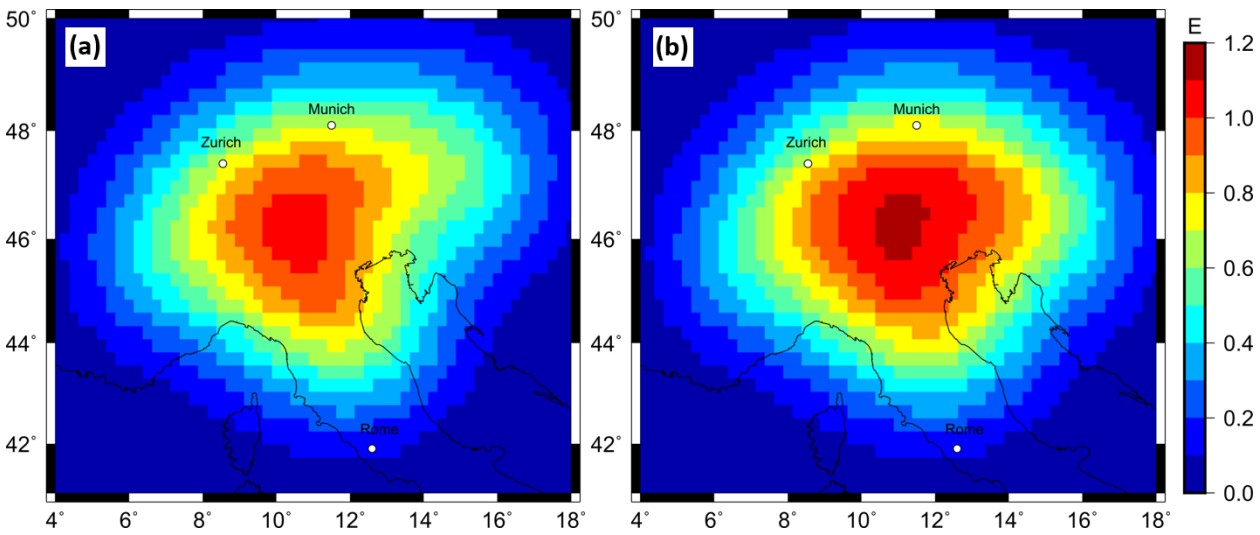


*Figure 15 Forward calculated gzz gravity signal at a station height of 225 km from predefined sub-crustal slab geometries with*
*a content density contrast of 60 kg/ m$^3$ and a constant thickness of 80 km. (a) slab configuration of hypothesis 1 (b) slab*
*configuration of hypothesis 2.*
In Addition, the signal strength for the forward calculated gzz component show the same
dependency of signal strength to the density contrast and slab thickness (Fig. 16) as the gz
component (Fig. 7). The signal strength of the gzz component ranges for the 12 different
combinations from 0.3 E to 2 E (Fig. 16). The gravity signal cannot be separated and affiliated
to a certain slab segment. The gzz gradient signal shows a large blurry gravity high over the
Alps, which thins out to the edges.

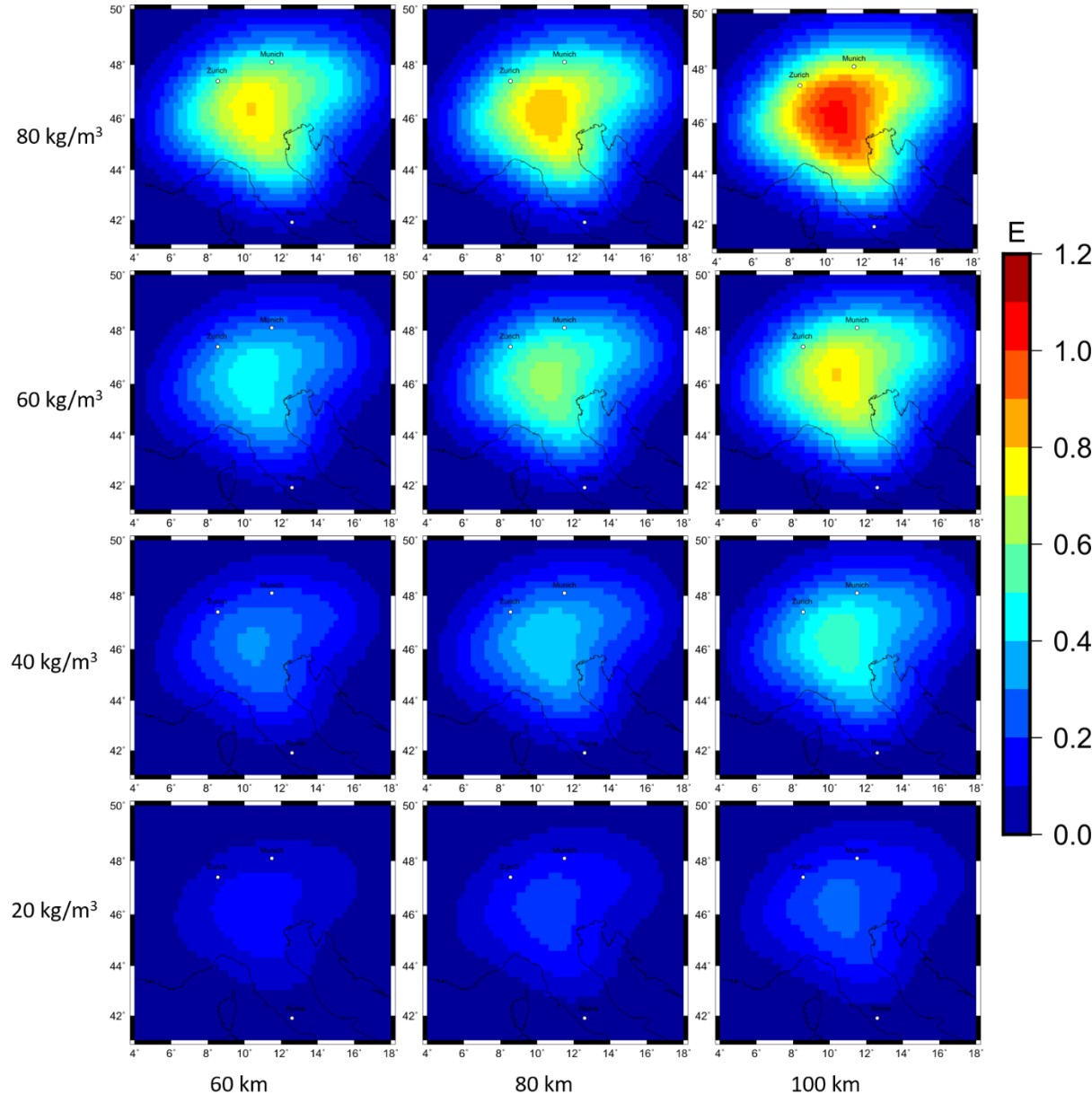

Figure 16 Forward calculated gzz gravity signal for 12 different combination of density contrast and slab thickness at a station height of 225 km for sub-crustal slab configuration 1.

The gravity effect for the LitMod models (section 4.2) with the slab Configuration 1 shows in the lithosphere domain a signal strength of about 0.05 E, while the sub lithospheric gravity signal is in the range of 0.1 E for the gzz component at satellite altitude of 225 km height. The combined gravity signal is in the order of 0.14 E (Fig. 17). A Proterozoic slab produces a larger amplitude in signal strength, however the different slab segments can again not be separated (Fig. 18).

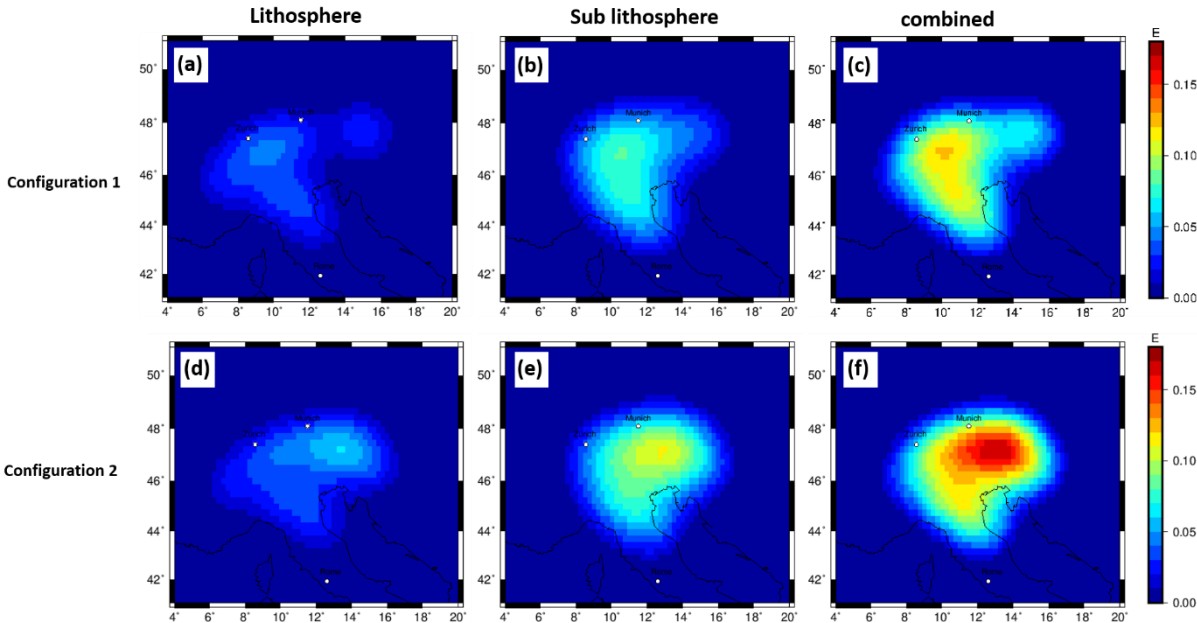

*Figure 17 forward calculated gzz gravity signal at satellite altitude of 225 km based on LitMod models with tecton like*
*compositions in the lithosphere and PUM and DMM compositions in the sub lithosphere ($M_1$, $M_2$, $M_3$, $M_4$) with an additional*
*thermal anomaly of -100° K for the sub-lithospheric slab segment, for predefined slab Configuration to the background model*
*$M_0$.* **(a)-(c)** *for Configuration 1.* **(d)-(f)** *for Configuration 2. Topographic and crustal effects are nullified.*

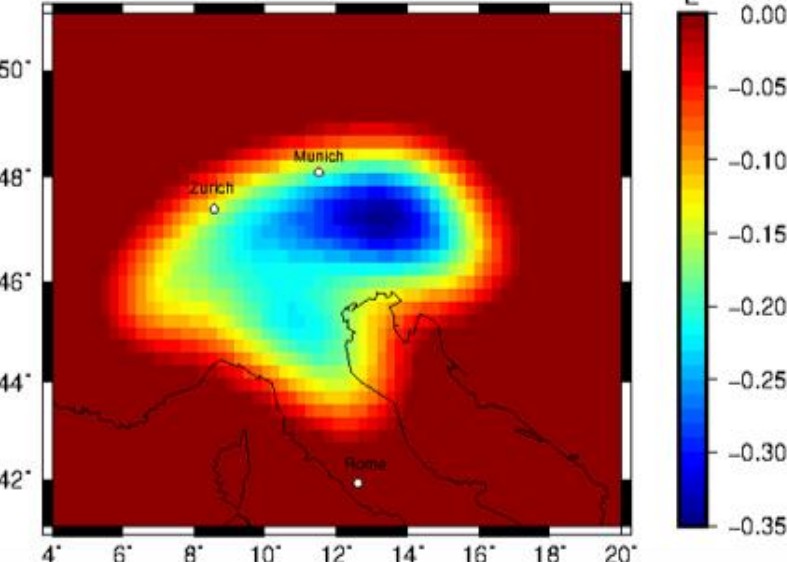

*Figure 18* Forward calculated gravity effect for the gzz component at satellite height of a Proterozoic lithospheric slab segment
to a Tecton compositional surrounding mantle for Configuration 2 obtained by calculating the residual between $M_8$ and $M_0$.