# Peer review of "Gravity Effect of Alpine Slab Segments Based on Geophysical and Petrological Modelling"

_Solid Earth, 2020_

## Referee Comment (RC1) · Carla Braitenberg (Referee) · 26 Oct 2020

**General Comments**

The study can be described a sensitivity analysis of the gravity field to the density variations in the depth range between 70 and 200 km below the Alps and surrounding areas. Several experiments are made to define the expected density variations. First a tomography model of percentage velocity variations respect to a reference model is converted with a constant conversion factor to expected densities contrasts respect to background model, leading to a density range +- 350 kg/m3. The corresponding gravity field extends over a range of 400 mGal, dominated by long wavelengths of several hundreds of km. Then the geometry of a number of lithospheric slabs is assumed to be present

below the Alps, and the possible slab geometries are defined from a selection of seismic tomography models. The depth range which is modelled goes from 40 to 220 km, and a fixed density contrast against the mantle of the slabs is defined. The thickness of the slabs varies between 60 and 100 km, density varies between 20 and 80 kg/m3, and the signal varies from a minimum of 30 mGal to a maximum of 140 mGal, depending on the density contrast and assumed slab thickness, with greater signal for thicker slab and greater density contrast. The effect of composition, temperature and pressure are considered when calculating the probable densities through the Litmod software. Essentially two mantle compositions, the Tecton and the Proterozoic type compositions are used for the two-layer background model of the mantle and for the subducting slabs, in different combinations of lithospheric mantle and sub-lithospheric mantle. The reference model has a lithosphere 100km thick, overlying the sub-lithospheric mantle, which should be equivalent to the asthenosphere. Here the slab is divided into a lithospheric and sub-lithospheric slab. Conceptually this is strange, since the subducting slab is made of lithosphere, and the asthenosphere would not rigidly participate in the subduction process. In the study, the subducting slab is divided into a lithospheric slab and a sub-lithospheric slab segment-with different compositions, and in some cases a reduction in temperature in the sub-lithospheric slab. I wonder whether this distinction is necessary, if it would not be sufficient to define a subducting slab of composition of different types, against the mantle reference model. In the Litmod modeling of the slabs, observing the vertical section of the model (Figs. 8 and 12), the slabs seem to be vertical columns, extending from the Moho down to over 200 km depth. Since in the first part of the study the slab geometries were defined through the seismic tomography. I wonder whether also the same inclined slab geometries were used in the Litmod model- from the figures it is not clear.

In Fig. 1 the simple Bouguer anomaly is shown in map, but the field is not used in the remainder of the manuscript, not in the comparison to the simulated fields nor in defining the residuals with the simulated fields, nor in the discussion. The estimation of the correctness of the simulated fields requires comparison with the observed gravity
field, which is presently lacking in the study. The only statement made, is that the observed amplitude of 200 mGal is not too much larger than the modelled slab effect of 40 mGal. Alternatively, the manuscript could clearly state that it is a sensitivity analysis to slab geometries through the gravity field in the Alpine area, without the target of formulation a realistic density model, since the verification with the observed field is lacking.

The sensitivity analysis as such is of value and is potentially of interest to scientists interested in defining slab geometries through geophysical modeling.

**Detailed comments**

Here first the text from the manuscript is copied, and comments start with the symbol "->"

L.10: The opposing slab configurations. -> This sentence is disconnected from previous one- better introduce the opposing slabs before.

L. 12: reflects or results? L.14-15: Therefore, we define the geometry of the upper slab interface by using the crustal thickness at 40 km depth as upper starting point -> This sentence is difficult to understand and the picture is not clear. If the slab is not resolved, why this would lead to defining upper slab interface? Starting point for what? Please reformulate abstract.

L. 16: the slab interface. -> Not clear which interface of slab is meant.

L. 17: -> the slab configuration is defined by the tomography or by the gravity modeling? Make it clear what the focus of the study is. The models consist in different assumed densities or also in slab geometries? If you mention configuration the reader expects different geometries, but above you claim you cannot resolve the geometry.

->I think the abstract should be reformulated to make the focus of the study clear from the beginning, and mention the given starting configuration. From the abstract it is not clear if the study contributes to the improvement of knowledge in slab geometry. It
seems the study is rather a sensitivity analysis, without conclusions on the real density structure from gravity modeling. This should be made clearer.

L. 42-43: In the Western Alps, Lippitsch et al. (2003) propose a slab break-off, which is in line with the findings of Beller et al. (2018) and Kästle et al. (2018)... -> Mention assumed depth of break-off.

L. 66: -> maybe you could add Tadiello and Braitenberg 2020, discussion paper.

L. 67 subduction-> subduction. ....dominated -> Crustal thickness variations and....

L. 68: subducting slabs have a higher density.. -> Give reference and justify. Is this always? Can it depend on lithospheric mantle composition and on the amount of subducting crustal thickness? What is ambient mantle- can you be more specific? For instance, below in your manuscript the density difference results to be either positive or negative.

-> Fig.1: red faults hardly seen in figure. For Bouguer anomaly specify topographic reduction chosen and maximum degree of the model.

L. 75: strength-> strengths

L. 75: Hereby, we convert seismic velocities to density. We also use seismic crustal thickness estimations and upper mantle tomographic models to define slab geometries -> The "also" is misleading- if not tomography what else can you use for the seismic velocities?

L. 83- -> give more details on how you make the correction of topography- how do you exactly calculate the simple Bouguer correction? Justify why you do not use the available complete mass effect of topography, which is available from the ICGEM homepage. Estimate what error you introduce by using simple Bouguer against the realistic mass correction. You mention the Etopo1 model with 1 minute resolution, how do you equalize frequency content in the topography mass correction and the gravity disturbance model which has a lower spatial resolution of 25 km? SED
L. 86 constant station height of 6040 m. -> Above geoid or above ellispoid or above a sphere? L. 88 indicating an isostatic compensation -> Compensation of what? Explain a bit better to make it clearer for non-specialists.

L. 89 gradients at 225 km height- -> justify why you also calculate gradients, and why you calculate them at a height different than gravity disturbance. Mention if you correct the mass effect on gradients.

L. 93: crustal thickness estimates based on the receiver function study by Spada et al. (2013), supplemented by the Moho depth model of the European plate by Grad et al. (2009) -> Show in fig.2 which areas come from Grad et al., and which from Spada, and which areas you have overlapping data-values. Do the two models agree in depth? At the western border of Po basin the Moho is shallow as in the Tyrrhenian sea, but this is not reflected in the gravity field. May there be a problem in the definition of Moho here?

L. 105 in the depth range from 70 to 200 km are calculated with respect to a depthdependent average shear-wave velocity 1-D model - Explain how you deal with the layers from 70 km depth to the Moho, as you mention that you correct gravity for the crust effects, that is from surface to the Moho. - Explain the values you choose for the 1D reference model and justify it. Does a different choice affect the results and in which respect? Add this point to the text. - Define relative velocity variations- I assume these are percentage values?

L. 125: relative densities -> Please make it clear what you intend with relative densitiese.g. density differences with respect to the reference density model divided by the density of the reference model?. Which is the reference model and how do you define it? Please specify. And as above, discuss the effect of a particular choice of the model. -> the conversion factor is a simplification of reality- mention which are the limits of this assumption. A linear function would be more general, that is delta\_rho=a+b delta \_V. You imply a=0, which probably is a parameter absorbed in the reference model. Please add some comments in the text to make it clear. SED
L. 126: conversion factor: from 0.1 to 0.45, as it is adimensional, the relative density and velocity is expressed in percentage? Make it clear in the text- see above comment.

L. 130: converted relative density distribution varies between -240 and 350 kg/m3. -> See previous comment: the relative density variation is not in percentage values, so the conversion factor must be expressed in correct units. Please make the text consistent.

L.131: High correlations between the structural pattern in the converted density distribution and the relative seismic velocities are observed -> This is obvious, as you imposed the linear relation.

L. 132: The converted 3D relative density distribution includes all heterogeneities in the Alpine lithosphere and not only structures of the potential slab segments to which the tomography is sensitive -> This is not clear- since the calculated density variation is just the seismic velocity variation multiplied by a factor, it represents exactly the variations sensed by seismic tomography, not more, not less. L. 142 In the gravity field, -> specify that you mean the modelled field or the observed field

L. 144 graity -> gravity -> Figure 4: at this point it would be interesting to learn how close this modelled field is to the observed field cleaned from the crustal contribution and the masses above 70 km depth. If the field cleaned from the crustal contribution is unavailable, you could maybe compare the modelled field with a low-pass filtered gravity field, or a field to which you have subtracted the greater crustal effects as those from the crustal thickness variation, for which you showed the model in Fig. 2.

L. 162 We define two alternative slab configurations based on a model of crustal thickness and different tomographic studies -> Two alternative configurations from two tomographic studies or several alternative configurations from several tomographic results? Please make it clear.

L. 163: Increasing crustal thickness is used as a direct indication for a subducting slab. -> Make this concept clearer. This is a model you use based on what assumptions?

SED
Can I have crustal thickness increase without a subducting slab? Make your concept clear. -> In Fig. 5 would it make sense to show the Moho 44 km isoline contour which defines the onset of the slab?

L. 168 -> I have a question on nomenclature of slab front- Is the slab front intended as the top boundary of the subducting slab and could this be a valid alternative to name it to make the picture clearer? Furthermore, I do not understand what you mean with vertical interpolation of isolines- I would understand the interpolation of the isolines defining the upper boundary of the slab to define its upper continuous surface- would this be correct? In Line 168 you define the slab front isoline at 200 km- so this would be the slab front, as the extreme of the slab. But if this is so, I find the term slab front for the upper boundary misleading. In Line 172 you mention the lower boundary of the slab, so the counterpart would be the upper boundary? L. 182: features a north subducting slab segment in the Eastern Alps -> Is it not NE-directed subduction?

L. 184: Central Alpine slab subducting in southeast direction -> rather SSE directed?

L. 185: southeast directed western alps slab -> rather ESE directed subduction?

L. 187: supporting the idea of slab break-off at about 100 km depth -> is the broken off segment still present or has it been absorbed and no velocity and density variations are seen?

L. 199: tesseroids modeling the slabs extend from 40 km to 220 km depth -> justify choice, as you say above that the slab starts at 44 km depth and extends to 200 km depth?

- L. 201 constraint -> constrained
- L. 215 signal -> signals
- L. 224- constant slab volume of 100 km, -> you mean slab thickness?
- L. 245 the perplex algorithm by -> the perplex algorithm of
**L. 251 homogene -> homogeneous**

L. 263 fix -> fixed

L. 270 Slab segments are introduced stepwise for the lithosphere and sub lithosphere domains into the model as well as thermal anomalies for the sub lithospheric model part -> Clarify at this point if the slabs you introduce have a different temperature than the surrounding lithosphere and sub-lithosphere mantle. The temperature anomaly is in the slab but only in the part of the slab that dips into the sub-lithospheric mantle? Please make it clearer in the text.

L. 272 a slab segments -> a slab segment

L. 283: Additional to the density contrast within the sub lithosphere, a temperature anomaly of – 100 K is introduced for the sub lithospheric part. -> Please make it clear if this last sentence refers to slab or surrounding mantle. -> The density variation is a consequence of the composition, temperature and pressure, so it is not clear why you mention that you introduced a density variation and a temperature variation- is the density variation not dependent on temperature?

L. 313- -> Fig. 10b. Define in caption which model it is (in text you mention M9). Slab is limited to Technical LAB depth? Make it clear in caption.

L. 317 - at surface height -> on topographic surface level?

L. 322, forward calculated gz gravity signal at...-> forward calculated gz gravity signal of Lithospheric slabs at...-> Make it clear in the text and maybe also in the caption that the gravity effect of topography and crustal thickness variations have been nulled.

L. 326 is in the order of 40 mGal -> is in the order of -40 mGal Fig. 13 A -> should be Fig. 12 A,

L. 334 Fig. 1 b -> should be Fig. 12 b

L. 335 Fig. 13 c -> should be Fig. 12 c,
L. 338 gravity response within the gravity field caused by the density distribution -> gravity response caused by the density distribution ?

L. 363- check grammar of sentence.

L. 364: as gravity modelling is a non-unique solution (Fig. 7). -> the forward model is exact and has one unique solution- the point is, that there is a tradeoff between density and volume, and the same gravity signal can be achieved with different combinations of density and geometry. Please change the sentence.

L. 368 In case, the... -> In the case that...

- L. 372 mantel -> mantle
- L. 377 from the the -> from the

L. 399 If the slab contribution is not considered, a significant part of the gravity field is attributed to crustal thickness or intra-crustal sources. -> This sentence is misleading, since it is not the identification of a specific geometry of one or more slabs that contributes to the signal, but in general the density variations in the depth range between 70 and 200 km, if identified as slabs or not. For instance, in the density model of the mantle of Fig. 3, the slabs have not been identified as such, but the mantle has a variable density. Therefore, it would be more correct to write that the subcrustal density variations contribute to the observed Bouguer field to an amount which is non-neglectable when modelling the crustal densities.

L. 424 All three modelling approaches suggest a positive gravitational effect of the Alpine slab segments up to 40 mGal -> from the gravity field of the first model, that converts the seismic model to density variations, the slab signal cannot be really identified. Maybe a positive modelled gravity is seen above the positive density variations in the center of the Figure, but amplitude with respect to the surroundings is much higher, more like 75 mGal. (taking difference from yellow isolines of -50 mGal to light red color of about +25 mGal).
L. 425 Previous studies compensated this effect by lithosphere thickness and/or intracrustal sources, future studies should incorporate the structures in order to provide a meaningful representation of the geodynamic complex Alpine area -> As far as I can see, other studies concerned with the crustal modeling have taken the sub-crustal density variations into account of the mantle, so it is not a big surprise that the density model must take into account both mantle and crust to reproduce the observations correctly. It could be mentioned at this point in the text that small scale density variations in the crust generate different wavelengths in the gravity field than deep slabs. Furthermore, it could be mentioned that a significant conclusion on the slab density structure requires a correct crustal density and crustal thickness model.

---

## Referee Comment (RC2) · Anonymous Referee #2 · 4 Nov 2020

The manuscript links seismology, petrology and gravity to discuss the effect of Alpine slabs on gravity anomalies. The petrological data mainly exploits composition to derive rock density information used in the modelling of two key scenarii: constant density distribution, and compositional + thermal variations with depth. Various other parameter values are varied to estimate the slabs' gravimetric signal. Beyond gravity anomaly calculations, gravity gradients are also presented but their significance is a question.

Overall, the calculations and motivation are interesting, but the goal, why and how this is done could be much better emphasized, even if the results are not conclusive in imaging slabs, but in estimating the amplitude of slabs' contribution towards the total gravity field. I'm still hesitant whether the initial goal of this work was to reproduce some features of the Alps, or to provide order-of-magnitude effects of various model

assumptions – probably the latter, but this could be more clearly expressed.

There are, in my reading, a number of points to clarify in the motivation, approach and discussion of the paper, as many elements remain less described than optimal. At the same time, goals could be better expressed from the beginning, as when the reader arrives to section 4, there is already some confusion on why and how certain steps were done, and what comes next. I try to highlight possible improvement points in my comments below.

There are also a number of moderate to major concerns regarding the methodology or choices that require, in my opinion, more than polishing the text. I also describe these here below.

Moderate to major concerns.

1) Seismic tomographies

Various results from seismic tomographies are mentioned in the introduction, but some significant papers are not included. Namely, numerous papers by the Prague group, discussing Eastern Alpine slab structures and proposed dual origin about 30 years ago. Since this is one of the main assumed models, I recommend these references are included and thoroughly cited in the manuscript. See also the EASI profile's receiver function results put in context of tomographic results in the Eastern Alps.

Moreover, the reader is referred to the paper by Kästle et al. 2020 for further discussion of Alpine tomographic models. I find that part of that publication is misleading as their Figure 3 juxtaposes different tomographic model pieces as if this was an accepted approach, while it is certainly not. Their Figure 3 has no colour scale – caption refers to Figure 3 itself – one can guess it is meant to refer to Figure 2, which shows that colour scales vary from tomography result to tomography result (5, 3, 4 % in VP, then 2.5 % in VS, from columns 1 to 4). Not only do these tomographies differ in the scale of shown anomaly amplitudes, but they also differ in the amount of data, their coverage,

their resolution, and inversion details. Therefore, for non-specialists, it is misleading to refer to this work as the Alpine tomography reference.

This is indeed a critical point as the model setup later on in the manuscript (e.g. Fig. 5) is also based on several tomographic sources, and is therefore subject to the inherent variability between tomographic models and their resolution. Errors stemming from merging several sources, or smoothed geometries across various choices, must be estimated in order to check their effects on the final results.

2) Velocity to density conversion

In section 2 of the manuscript, it is not always clear which kind of velocities are or will be used, only S or also P, and in which way. Please adjust the text to make this clear.

In section 3, a conversion factor is introduced that allows to convert seismic velocities to density. Although there is a list of references for the range of values, it would be beneficial to know whether this refers to P or S wave velocity conversion factors. Moreover, it would be very useful to see a formula for the conversion, how this is used, what are the units, does it refer to absolute or relative numbers? The numbers in the current manuscript leave me hesitant about this. Is there a T and P dependence of this conversion factor? Is it linear with no offset at zero? (I.e.: is the form y=a*x or y=a*x+b?) Why is the choice of 0.3 is taken in this study, what are the uncertainties of this choice?

3) First model setup and calculations

In section 3 some clarification is needed to resolve the followings.

Line133 says: "The converted 3D relative density distribution includes all heterogeneities in the Alpine lithosphere", but it is not clear how the depth range 0-70 km (the bulk of the lithosphere, and the most influential for the gravity values) is converted, or constructed in terms of density values. This is VERY critical in my opinion to be able to compare synthetics with observations.

Line142 says: "In the gravity field, a gravity high with a magnitude of ∼40 mGal is

observed over the Alps." This is not what is shown in the data. Later, I see this approaches what is on Figure 4, but there is no mention here about the forward model calculations, how they were done, how comparable are the results to the observed data. Why is this 40 mGal if the data showed a negative anomaly approaching -200 mGal (Figure 1). Is this related to the model definition between 0-70 km depths?

(Or, if the goal is not to compare it to observation, but simply to give estimates of slabs' gravimetric effects, then this needs to be clearly stated – and in this case it is not clear why gravity data is presented in Figure 1?)

4) Slab definitions

It seems that the slab volume (its definition) can be debated. Line 165 mentions 44 km (abstract and L199 refer to 40 km?) depth as the beginning of slabs. Why was this value chosen, and more importantly, what uncertainty do the RF Moho uncertainties represent in terms for slab volume? Then, Line 168 says that the 0% tomographic isoline was chosen to define a slab shape. Same question as above: why this value as chose? It seems very "optimistic" to call a slab anything that has fast, even marginally fast velocities. For example, is +0.01% anomaly part of a slab? Or +0.1%? Or can it be shown on cross-sections that this choice has a negligible consequence? Finally, on line 174, please specify how the thickness parameter was chosen (or was it inverted for?).

5) LitMod models

There are a few elements of LitMod model definitions that would be worth better arguing, or at least describing.

Line 250 and around: this is too succinct, please discuss why these model compositions were chosen (how were those models assembled, do they refer to continental material?), and how well they represent the Alps. See also around Line280-282 (e.g., why is a Proterozoic slab composition selected?).

Line 263 and around: choosing the LAB to define models is a different way that what was presented for constant-density models. Are the model results going to be comparable?

In particular, it is quite surprising to read about the constant LAB depth choice as a reference. Sparse data and that it is in discussion is a weak argument not to consider those data – the situation is the same for tomographic models. And if Artemieva's model shows +/-20 km variations in depth, why not include those?

Line 284 mentions a -100°K temperature anomaly. Is it for the slab? (Sentence ending on L285 says sub-litho.). Is this anomaly kept constant across the entire slab? How reasonable this is compared to the geodynamics of the Alps? Why aren't the thermal equations resolved if this is said to be an option in LitMod? Subsequently, models 7 and 8 are described to do include temperature variations (in space? I assume...), but Table 2 says T-anomaly of -200°K. What is the situation, then?

Figure 8 shows vertical slabs, and with different thicknesses. Why these two choices?

6) Discussion

The discussion is a mostly fair description of the results, stating some of the difficulties, limitations, and unresolved elements. Yet there are a few statements that can be contested:

L387 says: "For all three modelling approaches (section 3) a measurable gravity effect of the subducting slab segments is seen". I agree that all models produce gravity anomalies that are measurable. But I have doubts that this signals are distinguishable within the total field, when one considers field observations. This is even more critical as the anomaly levels vary a lot between constant-density and LitMod-based models. (Moreover, section 3 is not about modelling approaches, is that number a typo?)

L388 says: "The independent slab segments are distinguishable to a certain degree". I think this statement is over-interpreting the results. If the images would be shown

to someone not familiar with deep structure of the Alps suggested from seismology, I have doubts whether that person would point to independent slab segments. See also previous comment.

L390-397: here the comparison is between constant-density and LitMod model results, but the way these models were defined (tomography contours from 70 km depth, resp. from LAB at 100 km depth) raises questions on comparability.

L401-407: here is a clearer message on why these calculations are useful. Maybe consider taking this into the intro?

L415-417: the sharpness is pre-defined in the models, especially for temperature (which affects densities), and reality is probably less sharp than this.

7) Writing

Although the message of the paper can be mostly followed and understood, there is some variability in the level of information and in being "to-the-point". A proofreading after revision could smooth these out. Information pertaining to the same topic are sometimes found in rather different parts of the manuscript, these could be better grouped. An example: which method is used to calculate synthetic gravity anomalies, and what kind (how many) different model resolutions have been used. L135 says $0.2°$ * 3 km but does not mention the method, L197 mentioned the method but has $0.2°$ * 20 km. Many statements say slabs extend to 200 km depth, L199 says 220 km. It is a bit difficult to keep up with these information bits.

Minor comments on text that I have not really or well understood.

-Line14-15: unclear what is "crustal thickness at 40 km depth" – please resolve this oxymoron.

-I find the abstract could be more specific and to-the-point. Which kind of gravity data is used and how to constrain the models? What are those significant pattern differences, and which model seems more realistic?

-L31-33: revise unclear sentence: "A major role... plays the Adriatic microplate"

-L34: Adria's rotation as seen by GPS could be cited here.

-L50: before mentioning E. Alps, maybe mention the size of the potential slab gap between C. and E. Alps?

-L51: correct "is been"

-L51-54: briefly explain on which observables this "classical" view is based; and, then, which were the arguments for challenging these.

-L59: dual subduction was first proposed by Prague group, in the early 1990's, and not by these two very recent papers. Several publications from the Prague group discuss this in detail, please mention their interpretation of a dual subduction, possibly in chronological order (before Lippitsch).

-L67: correct "dominate"

-L66 and Fig. 1a: a much clearer description of what the Bouguer anomaly map is needed. What is XGM 2019? How was it obtained? What kind of resolution to expect? Or refer to what is written below. On Fig. 1b please make the fault lines more apparent (thicker line, or another colour).

-L74-79: a better and more detailed description of the motivation, approach and goals would be very welcome here. For example, petrology is not mentioned here at all.

-L81-83: please add how this model was obtained. Satellite (which satellite?) data only? Or also land data? To make it resolved at 25 km, what assumptions were taken? Local isostatic equilibrium of the topography? (If yes, is it surprising to see isostatic equilibrium in line 88?)

-L90: use of GOCE should be mentioned earlier. What was the purpose of using these gradients?

-L101: correct the reference year

-L105: how is the model constructed between Moho depth and 70 km depth level?

-L106: "depth-dependent average shear-wave velocity 1-D model" – is there some redundancy in the description?

-L108-111: the choice of 200 km depth as bottom of the model, and that anomalies below this level, seems ad hoc. Would it be possible to quantify/to justify? The El-Sharkawy et al. results extend to 300 km depth.

-L117: see earlier comment on dual subduction and its references.

-L139: the label says Vsv is used, is Vsh also available? Why choosing one rather than the other?

-L165: Please correct "crustal mantle boundary".

-L192 and 198: what is the resolution of the MeRE2020 tomographic model used here, and how does it compare to 0.2° horizontal and 20 km vertical tesseroid size?

-L197: finally comes the calculation details. I think this deserves defining a separate sub-section.

-L206: is it density anomaly instead of density?

-L210: uncertainty is mentioned for the first time. I agree this is important and I'm happy to see this word here. But could you please refer to uncertainty of gravity, models, tomographies also earlier in the manuscript, so that it does not come as a surprise here, and we know why parameter values were chosen as chosen?

-L211: it seems that a new section starts here, with results? If yes, please clearly state it (new sub-section). For the first model result presented here (drho=60kg/m3, Hslab=80km): why was this particular model chosen?

-L222: is it thickness rather than volume?

-L245 (twice): correct spelling Perple_X.

-L246: this is not what Perple_X does, this is one thing Perple_X can do. The choice of input (here 6 oxides only), the thermodynamic databases, list of solid solutions, etc. are all user chosen. Maybe the description is simply how LitMod uses Perple_X?

-L251: correct homogene to homogeneous, sub to sub-

-L275: this could be presented much more nicely, with several columns representing the type of information (lithosphere, sub-lithosphere, slab config., T anomaly, etc.) and the lines showing the information itself. From the current version it is very difficult to have an overview of tested models.

-L295: is this the beginning of results? If yes, mark it with a sub-section, for example.

-L306: please clarify what causes these <1kg/m3 variations, temperature?

-Figure 11: please match "Configuration" in caption and "Hypothesis" in the figure.

-L372: typo in mantel

-L386: how realistic is this -100°K anomaly, and, therefore, the 16 mGal anomaly?

-In the conclusions, I'd recommend mentioning that future results based on AlpArray Seismic Network data will be of high interest in better defining slab geometries AND properties.

Appendix

-L634-635: some references to the data would be useful.

-L637: correct "longwave length"

-L641-645: are the obtained results in E on the same order of magnitude that you would expect? What do these maps mean, then? What is the support for the interpretation on lines 650-652 (positive signal of 0.5 E could be linked to slabs)?

-Figure 14: please use the same colour scale and range as on Figure 13.

[Figure]

---

## Author Comment (AC1) · 17 Dec 2020

**Ref: se-2020-145**

**Title: Gravity Effect of Alpine Slab Segments Based on Geophysical and Petrological Modelling" by Maximilian Lowe et al.**

We revised our manuscript in line with the in-depth reviews made by Carla Braitenberg (RC1) and an anonymous referee (RC2). We thank the referees for the constructive criticism and suggestions, which helped us to improve the manuscript.

Both reviews made it clear, that we have to express our aims more clearly and that regrouping of some elements is needed. Accordingly, we reformulated the abstract and introduction with a strong emphasis on introducing our three different modelling approaches as well as the aim to investigate gravity sensitivity to subducting slab segments. Moreover, we separate the conclusion regarding the three modelling approaches, so that they match the new introduction better.

We hope that the new version of the manuscript is improved and can be accepted for publication.

All points raised by both reviewers have been addressed. Response to reviewer 1 from page 1 to 9. Response to reviewer 2 from page 9 to 17.

**Carla Braitenberg Referee # 1**

***General Comments***

*The study can be described a sensitivity analysis of the gravity field to the density variations in the depth range between 70 and 200 km below the Alps and surrounding areas. Several experiments are made to define the expected density variations.*

*First a tomography model of percentage velocity variations respect to a reference model is converted with a constant conversion factor to expected densities contrasts respect to background model, leading to a density range +- 350 kg/m3.*

For clarification: we use the absolute values to calculate a percentage-based differences between densities and seismic velocities. We include now references to the linear relationship between seismic velocities and densities (Tiberi et al. 2001; Webb 2009) and reformulated this paragraph (section 3) to describe our approach in more detail.

*The corresponding gravity field extends over a range of 400 mGal, dominated by long wavelengths of several hundreds of km. Then the geometry of a number of lithospheric slabs is assumed to be present below the Alps, and the possible slab geometries are defined from a selection of seismic tomography models. The depth range which is modelled goes from 40 to 220 km, and a fixed density contrast against the mantle of the slabs is defined. The thickness of the slabs varies between 60 and 100 km, density varies between 20 and 80 kg/m3, and the signal varies from a minimum of 30 mGal to a maximum of 140 mGal, depending on the density contrast and assumed slab thickness, with greater signal for thicker slab and greater density contrast.*

*The effect of composition, temperature and pressure are considered when calculating the probable densities through the Litmod software. Essentially two mantle compositions, the Tecton and the Proterozoic type compositions are used for the two-layer background model of the mantle and for the subducting slabs, in different combinations of lithospheric mantle and sub-lithospheric mantle. The reference model has a lithosphere 100km thick, overlying the sub-lithospheric mantle, which should be equivalent to the asthenosphere. Here the slab is divided into a lithospheric and sub-lithospheric slab. Conceptually this is strange, since the subducting slab is made of lithosphere, and the asthenosphere would not rigidly participate in the subduction process.*

The LAB is defined in LitMod as the 1300°C isotherm. Therefore, the division of the slab segments is introduced to avoid that this thermal isoline is following the shape of the slab segments and effectively would sit beneath the slab at 200 km depth (maximum depth of

modelled slab segments). The division in lithospheric and sub-lithospheric slab segment is based on the thermal definition in LitMod. We follow here the approach and model discretisation from Fullea et al. 2015 where the authors model subducting slab segments in the Atlantic-Mediterranean Transition Region. We edited the corresponding paragraph in the manuscript (section 4.2) to make this decision in the model discretisation clearer.

*In the study, the subducting slab is divided into a lithospheric slab and a sub-lithospheric slab segment-with different compositions, and in some cases a reduction in temperature in the sub-lithospheric slab. I wonder whether this distinction is necessary, if it would not be sufficient to define a subducting slab of composition of different types, against the mantle reference model.* We follow Fullea et al. 2015 with a similar differentiation. See also our responses to previous comments. We address this now in the manuscript to avoid confusion.

*In the Litmod modeling of the slabs, observing the vertical section of the model (Figs. 8 and 12), the slabs seem to be vertical columns, extending from the Moho down to over 200 km depth. Since in the first part of the study the slab geometries were defined through the seismic tomography, I wonder whether also the same inclined slab geometries were used in the Litmod model- from the figures it is not clear.*
We use in fact two different input models regarding the subduction angle for the Tesseroid and the LitMod models. For the Tesseroid model, we extract the geometry of the slab segments as well as the subduction angle of the slab segments from different tomography models, while for the LitMod models we extract the slab geometries at the depth of the Moho interface (identical isoline than for the tesseoid models) and assume a vertical extension down to 200 km depth. This difference in input slab models is due to the different discretization of the Software Tesseroids & LitMod 3D. In Tesseroid we define the slab volume in a 3-dimensional space, while in LitMod 3D we implement the slabs as 2.5 dimensional layers. Meaning we have a grid with x and y coordinates and a corresponding z value. We can therefore only use one z value for a given coordinate pair. Implementing a subduction zone with a subducting angle unequal 90 degrees is not possible without considerable effort.

*In Fig. 1 the simple Bouguer anomaly is shown in map, but the field is not used in the remainder of the manuscript, not in the comparison to the simulated fields nor in defining the residuals with the simulated fields, nor in the discussion. The estimation of the correctness of the simulated fields requires comparison with the observed gravity field, which is presently lacking in the study. The only statement made, is that the observed amplitude of 200 mGal is not too much larger than the modelled slab effect of 40 mGal. Alternatively, the manuscript could clearly state that it is a sensitivity analysis to slab geometries through the gravity field in the Alpine area, without the target of formulation a realistic density model, since the verification with the observed field is lacking.*
The Bouguer Anomaly map is used to illustrate the negative gravity signal, which remains after correcting for topography. We want to motivate our study by showing that the gravity signal of subducting slab segments, which can be assumed to be positive, is not visible in the Bouguer Anomaly Map and justifying therefore, the effort to model the gravity signal of the slab segments.
We do not compare our models to the observed Alpine gravity field mainly because, we do not account for crustal variations. The crustal thickness is the dominating contributor in the observed gravity field and the proportion of the crustal gravity field cannot be easily removed or filtered. We present here a sensitivity study of the gravity signal caused by subducting slab segments as stated in the second part of the comment by Carla Braitenberg. In the revised version of the manuscript, we make this clearer throughout the different section of manuscript (i.e. Abstract, Introduction, Conclusions). Comparison to the observed Alpine gravity field would require an integrated modelling of the slab segments, the surrounding mantle and the crust, which are future research directions. We make a clearer statement of this in the introduction and the result section.

*The sensitivity analysis as such is of value and is potentially of interest to scientists interested in defining slab geometries through geophysical modeling.*

**Detailed comments**
*L.10: The opposing slab configurations. -> This sentence is disconnected from previous one-better introduce the opposing slabs before.*
*L. 12: reflects or results? L.14-15: Therefore, we define the geometry of the upper slab interface by using the crustal thickness at 40 km depth as upper starting point -> This sentence is difficult to understand and the picture is not clear. If the slab is not resolved, why this would lead to defining upper slab interface? Starting point for what? Please reformulate abstract.*
*L. 16: the slab interface. -> Not clear which interface of slab is meant.*
*L. 17: -> the slab configuration is defined by the tomography or by the gravity modeling? Make it clear what the focus of the study is. The models consist in different assumed densities or also in slab geometries? If you mention configuration the reader expects different geometries, but above you claim you cannot resolve the geometry. ->I think the abstract should be reformulated to make the focus of the study clear from the beginning, and mention the given starting configuration. From the abstract it is not clear if the study contributes to the improvement of knowledge in slab geometry. It seems the study is rather a sensitivity analysis, without conclusions on the real density structure from gravity modeling. This should be made clearer.*
Comments from L10 - L17 are aimed at the abstract regarding the readability and structure. We performed a major reformulation of the Abstract, as suggested

*L. 42-43: In the Western Alps, Lippitsch et al. (2003) propose a slab break-off, which is in line with the findings of Beller et al. (2018) and Kästle et al. (2018): : : -> Mention assumed depth of break-off.*
The assumed slab break-off depth is in about 100 km depth. -> added to the manuscript at line 97-99.

*L. 66: -> maybe you could add Tadiello and Braitenberg 2020, discussion paper.*
Added

*L. 67 subduction-> subduction. ...dominated -> Crustal thickness variations and …*
*L. 68: subducting slabs have a higher density… -> Give reference and justify. Is this always? Can it depend on lithospheric mantle composition and on the amount of subducting crustal thickness? What is ambient mantle- can you be more specific? For instance, below in your manuscript the density difference results to be either positive or negative.*
The introduction has been reformulated, as a result the corresponding paragraph to this comment changed substantially. We added references to the statement that subducting slabs have a higher density compared to the surrounding mantle at identical depth levels. Ambient mantle was used as a synonym of surrounding mantle. In the updated manuscript we avoided using this synonym to avoid confusion. We create in fact a slab segment in the LitMod section, which has lower densities then the background model, due to the isopycnicity effect that Proterozoic material is older and therefore denser then Tecton composition but due to mineral depletion the composition counteract the thermal effect and therefore Proterozoic composition is in fact less dense then the Tecton composition, which is assigned to our background model. Here we aim to illustrate the effect composition has on the density distribution within the slab and to the surround mantle and show the importance of correct compositional information, therefore we focus on the difference in density contrast between slab and surrounding mantle and neglecting the sign of the density contrast. We state this clearer in the updated manuscript.

*-> Fig.1: red faults hardly seen in figure. For Bouguer anomaly specify topographic reduction chosen and maximum degree of the model.*
Line width increased, explanation of the key parameter for the Bouguer correction added to the caption. Changed the arrangement of the figure, topographic map is now on the left side, while the Bouguer Anomaly map is on the right.

*L. 75: strength-> strengths*
Corrected

*L. 75: Hereby, we convert seismic velocities to density. We also use seismic crustal thickness estimations and upper mantle tomographic models to define slab geometries -> The "also" is misleading- if not tomography what else can you use for the seismic velocities?*
Corrected

*L. 83- -> give more details on how you make the correction of topography- how do you exactly calculate the simple Bouguer correction? Justify why you do not use the available complete mass effect of topography, which is available from the ICGEM homepage. Estimate what error you introduce by using simple Bouguer against the realistic mass correction. You mention the Etopo1 model with 1 minute resolution, how do you equalize frequency content in the topography mass correction and the gravity disturbance model which has a lower spatial resolution of 25 km?*
We supplemented our explanation of the Bouguer Anomaly calculation in section 2. Following a short statement on the calculation: We calculated the gravity signal of the topography and bathymetry at a station height of 6040 m using Tesseroids with a correction density of 2670 kg/m$^3$ for the topography and a correction density for water of 1030 kg/m$^3$. The necessary height informations are taken from ETOPO which was regrided with a 25 km grid space. The resulting gravity signal was then subtracted from the Free-Air Anomaly in order to obtain the Bouguer Anomaly. An error estimation is not valuable to this study because we do not compare our findings to the observed gravity field, but rather use the Bouguer Anomaly to motivate our study, as stated in the last reply to the general comments (page 2).

*L. 86 constant station height of 6040 m. -> Above geoid or above ellipsoid or above a sphere?*
*L. 88 indicating an isostatic compensation -> Compensation of what? Explain a bit better to make it clearer for non-specialists.*
Added clarification that our stations are above the ellipsoid. A large negative Bouguer Anomaly indicates isostatic compensation of topographic load in from of a crustal root. We added clarification to the compensation of topography.

*L. 89 gradients at 225 km height- -> justify why you also calculate gradients, and why you calculate them at a height different than gravity disturbance. Mention if you correct the mass effect on gradients.*
Added clarification in section 2, that in fact we are correcting the gravity gradients measured by GOCE for topography as well.
We reformulate our introduction (section 1) to make our intention of the manuscript clearer from the start. Here, the justification of using gravity gradients at satellite height altitude is given. To summarize quickly, we anticipated that the gravity gradients at satellite altitude are sensitive to the slab segments. That is why we calculated gz just above the Alps surface and the gradients at GOCE altitude. Our finding shows that gravity gradients at satellite height are not sensitive to the different slab segments, and therefore we moved research corresponding to the gradients in the appendix. We believe that those findings are still significant to the manuscript and therefore, worthwhile keeping.

*L. 93: crustal thickness estimates based on the receiver function study by Spada et al. (2013), supplemented by the Moho depth model of the European plate by Grad et al. (2009) -> Show*

*in fig.2 which areas come from Grad et al., and which from Spada, and which areas you have overlapping data-values. Do the two models agree in depth? At the western border of Po basin the Moho is shallow as in the Tyrrhenian sea, but this is not reflected in the gravity field. May there be a problem in the definition of Moho here?*

We include now separated Figures showing the different Moho depth estimation maps as well as the merged product. The depth values are not identical of both crustal depth estimations, therefore a cosine taper is used to blend both grids using distance weighting. Important to note the major Alpine area is taken from Spada et al (2013), while only the edges are filled by Grad et al. (2009)

*L. 105 in the depth range from 70 to 200 km are calculated with respect to a depth dependent average shear-wave velocity 1-D model - Explain how you deal with the layers from 70 km depth to the Moho, as you mention that you correct gravity for the crust effects, that is from surface to the Moho. - Explain the values you choose for the 1D reference model and justify it. Does a different choice affect the results and in which respect? Add this point to the text. - Define relative velocity variations- I assume these are percentage values?*

We choose an upper limit of 70 km because: i) we want to remove the crust; ii) we want a uniform starting model for the converted density model (section 3); iii) above 70 km the slab segments in the tomographic model MeRE2020 are not well recovered / the model is not sensitive to shallow structures. We included this explanation in the corresponding paragraph.

*L. 125: relative densities -> Please make it clear what you intend with relative densities e.g. density differences with respect to the reference density model divided by the density of the reference model?. Which is the reference model and how do you define it? Please specify. And as above, discuss the effect of a particular choice of the model. -> the conversion factor is a simplification of reality- mention which are the limits of this assumption. A linear function would be more general, that is delta_rho=a+b delta _V. You imply a=0, which probably is a parameter absorbed in the reference model. Please add some comments in the text to make it clear.*

We added the formula to the manuscript and described more precise how the density distribution is converted from the seismic velocities. To quickly summarize: the percentage deviation of absolute velocity to the background model is estimated and multiplied by the conversion factor. Added reference to the manuscript describing the linear relation between seismic velocity variation and density variation (Tiberi et al. 2001 ; Webb 2009).

*L. 126: conversion factor: from 0.1 to 0.45, as it is adimensional, the relative density and velocity is expressed in percentage? Make it clear in the text- see above comment.*

See reply to comment L.125

*L. 130: converted relative density distribution varies between -240 and 350 kg/m3. -> See previous comment: the relative density variation is not in percentage values, so the conversion factor must be expressed in correct units. Please make the text consistent.*

See reply to comment L.125

*L.131: High correlations between the structural pattern in the converted density distribution and the relative seismic velocities are observed -> This is obvious, as you imposed the linear relation.*

Yes indeed, the similarity of the structural pattern is expected due to the linear relation. The similarity in Figure 3 is a proof of concept.

*L. 132: The converted 3D relative density distribution includes all heterogeneities in the Alpine lithosphere and not only structures of the potential slab segments to which the tomography is sensitive -> This is not clear- since the calculated density variation is just the seismic velocity variation multiplied by a factor, it represents exactly the variations sensed by seismic tomography, not more, not less.*

*Reformulate paragraph to remove any ambiguity and inaccurate formulations.*

*L142 In the gravity field, -> specify that you mean the modelled field or the observed field.*
corrected

*L. 144 graity -> gravity -> Figure 4: at this point it would be interesting to learn how close this modelled field is to the observed field cleaned from the crustal contribution and the masses above 70 km depth. If the field cleaned from the crustal contribution is unavailable, you could maybe compare the modelled field with a low-pass filtered gravity field, or a field to which you have subtracted the greater crustal effects as those from the crustal thickness variation, for which you showed the model in Fig. 2.*
In the Bouguer gravity field density contrast at the Moho is the dominating contribution, we also do not consider any masses below 200 km, which would contribute a long wavelength proportion. We believe using a high and low pass filter to isolate the depth interval (70km – 200km), which has a low gravity contribution as shown in our findings would result in biased gravity fields. To be able to compare to the observed gravity field we need an integrated crust and lithospheric models, which is beyond the scope of the present study.

*L. 162 We define two alternative slab configurations based on a model of crustal thickness and different tomographic studies -> Two alternative configurations from two tomographic studies or several alternative configurations from several tomographic results? Please make it clear.*
Added clarification that we in fact use several tomographic models, also added a remake that the precise description of the slab configuration can be found below in the manuscript.

*L. 163: Increasing crustal thickness is used as a direct indication for a subducting slab. -> Make this concept clearer. This is a model you use based on what assumptions? Can I have crustal thickness increase without a subducting slab? Make your concept clear. -> In Fig. 5 would it make sense to show the Moho 44 km isoline contour which defines the onset of the slab?*
Reformulated corresponding paragraph. Added Moho label to the Moho isoline in Figure 5.

*L. 168 -> I have a question on nomenclature of slab front- Is the slab front intended as the top boundary of the subducting slab and could this be a valid alternative to name it to make the picture clearer? Furthermore, I do not understand what you mean with vertical interpolation of isolines- I would understand the interpolation of the isolines defining the upper boundary of the slab to define its upper continuous surface- would this be correct? In Line 168 you define the slab front isoline at 200 km- so this would be the slab front, as the extreme of the slab. But if this is so, I find the term slab front for the upper boundary misleading. In Line 172 you mention the lower boundary of the slab, so the counterpart would be the upper boundary?*
We changed the nomenclature from slab front to upper boundary of the slab. As the comment implies it makes the text clearer and is less misleading.

*L. 182: features a north subducting slab segment in the Eastern Alps -> Is it not NE-directed subduction?*
corrected

*L. 184: Central Alpine slab subducting in southeast direction -> rather SSE directed?*
Corrected

*L. 185: southeast directed western alps slab -> rather ESE directed subduction?*
Corrected

*L. 187: supporting the idea of slab break-off at about 100 km depth -> is the broken off segment still present or has it been absorbed and no velocity and density variations are seen?*
We only considered attached slab segments here. We can do this because we only estimate the gravity signal of the slab segments and do not compare our results to the measured gravity signal. Therefore, mantle upwelling in the area of slab break off and or detached slab segments are not modelled here. More sophisticated and integrated gravity and or density models of the Alpine region needs to include the crust, lithosphere and such mechanisms as detachment and mantle upwelling in order to recreate realistic physical properties of the Alps. See also reply in the general comment section.

*L. 199: tesseroids modeling the slabs extend from 40 km to 220 km depth -> justify choice, as you say above that the slab starts at 44 km depth and extends to 200 km depth?*
The 220 km depth boundary originate from the way the tesseroid input file is created. Basically, I use a for loop over the different depth intervals with a 20 km increment. The loop runs over the depth interval from 40 to 200 km, that means, that at 200 km depth a last Tesseroid is defined with an upper boundary of 200 km and a lower boundary at 220 km. We could argue that we can do that since we clearly observe the slab at the 200 km depth slice in the MeRE2020 model. However, it introduces an unnecessary inconsistence to the two other presented approaches (Converted densities and LitMod). Therefore, we recalculated the tesseroid models with a maximum depth of 200 km and updated therefore figure 6,7,15 and 16 in the manuscript. For the tesseroid model with 60 kg/m3 density contrast and a slab thickness of 80 km (Figure 6 manuscript) the different between the new and the old calculation is ~ 4 mGal (See Figures below). Even though the new results do not change the meaning of the manuscript it is a significant change which is worth updating, which we did.

[Figure]

*Figure 1 **left)** old calculation from the original submitted manuscript. **Middle)** updated tesseroid model. **Right)** Residual between original tesseroid model and updated tesseroid model.*

Secondly, the tesseroids are defined at 40 km depth due to the 20km vertical expansion. The Tesseroids ranging from 40 to 60 km are corrected for the densities corresponding from 40 km to 44 km, as a result the tesseroids density corresponds to a tesseroid ranging from 44 to 60 km. Clarification added to the manuscript.

*L. 201 constraint -> constrained*
Corrected

*L. 215 signal -> signals*
Corrected

*L. 224- constant slab volume of 100 km, -> you mean slab thickness?*
Corrected

*L. 245 the perplex algorithm by -> the perplex algorithm of*
Corrected

*L. 251 homogene -> homogeneous*

Corrected

*L. 263 fix -> fixed*
Corrected

*L. 270 Slab segments are introduced stepwise for the lithosphere and sub lithosphere domains into the model as well as thermal anomalies for the sub lithospheric model part -> Clarify at this point if the slabs you introduce have a different temperature than the surrounding lithosphere and sub-lithosphere mantle. The temperature anomaly is in the slab but only in the part of the slab that dips into the sub-lithospheric mantle? Please make it clearer in the text.*
The thermal anomalies are added to the slab segments beneath the technical LAB, which describes the 1300-degree thermal isoline. Following the scheme after Fullea et al. 2015.

*L. 313- -> Fig. 10b. Define in caption which model it is (in text you mention M9). Slab is limited to Technical LAB depth? Make it clear in caption.*
Incorporated the suggestion above.

*L. 317 - at surface height -> on topographic surface level?*
Yes, at topographic level. Corrected in the manuscript.

*L. 322, forward calculated gz gravity signal at: : :-> forward calculated gz gravity signal of Lithospheric slabs at: : : -> Make it clear in the text and maybe also in the caption that the gravity effect of topography and crustal thickness variation have been nulled.*
Incorporated in the manuscript

*L. 326 is in the order of 40 mGal -> is in the order of -40 mGal Fig. 13 A -> should be Fig. 12 A,*
Corrected

*L. 334 Fig. 1 b -> should be Fig. 12 b*
Corrected

*L. 335 Fig. 13 c -> should be Fig. 12 c,*
Corrected

*L. 338 gravity response within the gravity field caused by the density distribution -> gravity response caused by the density distribution?*
corrected

*L. 363- check grammar of sentence.*
corrected

*L. 364: as gravity modelling is a non-unique solution (Fig. 7). -> the forward model is exact and has one unique solution- the point is, that there is a tradeoff between density and volume, and the same gravity signal can be achieved with different combinations of density and geometry. Please change the sentence.*
We changed the corresponding sentence and made our statement clearer.

*L. 368 In case, the: : : -> In the case that: : :*
Corrected

*L. 372 mantel -> mantle*
Corrected

*L. 377 from the the -> from the*

Corrected

*L. 399 If the slab contribution is not considered, a significant part of the gravity field is attributed to crustal thickness or intra-crustal sources. -> This sentence is misleading, since it is not the identification of a specific geometry of one or more slabs that contributes to the signal, but in general the density variations in the depth range between 70 and 200 km, if identified as slabs or not. For instance, in the density model of the mantle of Fig. 3, the slabs have not been identified as such, but the mantle has a variable density. Therefore, it would be more correct to write that the subcrustal density variations contribute to the observed Bouguer field to an amount which is nonneglectable when modelling the crustal densities.*
Reformulated corresponding paragraph.

*L. 424 All three modelling approaches suggest a positive gravitational effect of the Alpine slab segments up to 40 mGal -> from the gravity field of the first model, that converts the seismic model to density variations, the slab signal cannot be really identified. Maybe a positive modelled gravity is seen above the positive density variations in the center of the Figure, but amplitude with respect to the surroundings is much higher, more like 75 mGal. (taking difference from yellow isolines of -50 mGal to light red color of about +25 mGal).*
Reformulated corresponding sentence. Also removed the generalisation of all 3 approaches and grouped approach 2 & 3 together, while separating out approach 1

*L. 425 Previous studies compensated this effect by lithosphere thickness and/or intracrustal sources, future studies should incorporate the structures in order to provide a meaningful representation of the geodynamic complex Alpine area -> As far as I can see, other studies concerned with the crustal modeling have taken the sub-crustal density variations into account of the mantle, so it is not a big surprise that the density model must take into account both mantle and crust to reproduce the observations correctly. It could be mentioned at this point in the text that small scale density variations in the crust generate different wavelengths in the gravity field than deep slabs Furthermore, it could be mentioned that a significant conclusion on the slab density structure requires a correct crustal density and crustal density thickness model.*
We expanded our conclusion with the suggested point. We aimed for this statement in the original manuscript, however, it appears it was not clear enough.

**Anonymous Referee #2**
**Major comments**
*1) Seismic tomographies*
*Various results from seismic tomographies are mentioned in the introduction, but some significant papers are not included. Namely, numerous papers by the Prague group, discussing Eastern Alpine slab structures and proposed dual origin about 30 years ago. Since this is one of the main assumed models, I recommend these references are included and thoroughly cited in the manuscript. See also the EASI profile's receiver function results put in context of tomographic results in the Eastern Alps.*
We now mention the papers by Babuska et al. (1990); Karousova et al. (2013), and Hentenyi et al. (2018).

*Moreover, the reader is referred to the paper by Kästle et al. 2020 for further discussion of Alpine tomographic models. I find that part of that publication is misleading as their Figure 3 juxtaposes different tomographic model pieces as if this was an accepted approach, while it is certainly not. Their Figure 3 has no colour scale – caption refers to Figure 3 itself – one can guess it is meant to refer to Figure 2, which shows that colour scales vary from tomography result to tomography result (5, 3, 4 % in VP, then 2.5 % in VS, from columns 1 to 4). Not only do these tomographies differ in the scale of shown anomaly amplitudes, but they also differ in*

*the amount of data, their coverage their resolution, and inversion details. Therefore, for non-specialists, it is misleading to refer to this work as the Alpine tomography reference.*

Our study aims to estimate the gravity response of the density structure in subducting slab segments. The density contribution of subducting slab segments is sparsely studied. We choose the Alpine region because of a range of recent topographies. In the mentioned paragraph we tried to give an overview of tomographic studies which were carried out in the Alpine region. Also, this paragraph is intended to illustrate how different tomographic models contradict each other. Here we want to motivate to include additional geophysical observations e.g. gravity. The aim of this study is to quantify the sensitivity of gravity measurements to sub crustal density variation caused by subducting slabs in order to evaluate if gravity could be a useful geophysical observation to be included in the discussion of the slab geometries and slab properties.

In Kaestle et al. (2020) it is clear that different methods are used for different studies as well as different types of waves (surface waves, P and S waves). For the purpose of our study we are interested in different geometries of the slab segment as seen from different studies. Since we aim to show the differences in the gravity signal caused by a variation of the slab geometry, we are not dependent on consistent tomographic models or waves types. We choose slab configurations from a variety of tomographic models and use their information about the geometry of the slabs as an input. We construct two different slab configurations based on this approach in order to illustrate how the forward calculated gravity response changes accordingly. To make it clear, we do not claim that those two configurations correspond to the real Alpine slab configuration, and we also do not claim that those two configurations, which are more hypotheses, are the only two valid hypotheses for the Alpine region. We selected different slab segments, which are imaged by different tomographies to create two competing slab configurations in order to study the effect of the geometry and estimate the sensitive to which extend gravity modelling can separate different configurations.

*This is indeed a critical point as the model setup later on in the manuscript (e.g. Fig. 5) is also based on several tomographic sources, and is therefore subject to the inherent variability between tomographic models and their resolution. Errors stemming from merging several sources, or smoothed geometries across various choices, must be estimated in order to check their effects on the final results.*

We test the gravity response to different slab geometries (see reply above). However, we do not compare the modelled gravity response to the observed gravity field as we are interested in quantifying the sub-crustal gravitational effect of the subducting slab segments and therefore nullified the crust and topography in our models. Future studies, which aim to resolve the real physical properties of the Alpine subsurface need to integrate crustal and topographical models.

*In section 2 of the manuscript, it is not always clear which kind of velocities are or will be used, only S or also P, and in which way. Please adjust the text to make this clear. In section 3, a conversion factor is introduced that allows to convert seismic velocities to density. Although there is a list of references for the range of values, it would be beneficial to know whether this refers to P or S wave velocity conversion factors. Moreover, it would be very useful to see a formula for the conversion, how this is used, what are the units, does it refer to absolute or relative numbers? The numbers in the current manuscript leave me hesitant about this. Is there a T and P dependence of this conversion factor? Is it linear with no offset at zero? (I.e.: is the form y=a\*x or y=a\*x+b?) Why is the choice of 0.3 is taken in this study, what are the uncertainties of this choice?*

This comment is inline of the remarks made by Carla Braitenberg (Referee #1) and our reply applies here as well. We reformulated the corresponding paragraph regarding the conversion of densities based on seismic velocities, we include now the formula as well as references to the linear relationship between seismic velocities and densities (Tiberi et al. 2001; Webb 2009). From both reviewers we have the impression that it was not clear enough what our

goals are regarding the conversion of densities from seismic velocities. We motivate our goals more precise in the introduction, so that the reader is not supervised or confused when he/she arrives at this section in the manuscript.

*3) First model setup and calculations*
*In section 3 some clarification is needed to resolve the followings.*
*Line133 says: "The converted 3D relative density distribution includes all heterogeneities in the Alpine lithosphere", but it is not clear how the depth range 0-70 km (the bulk of the lithosphere, and the most influential for the gravity values) is converted, or constructed in terms of density values. This is VERY critical in my opinion to be able to compare synthetics with observations.*
We do not aim to compare the forward calculated gravity field to the observed gravity field because we neglected the contribution of the crust and topography, as stated above. In the response to Carla Braitenbergs comments to L 105, L 144 and L199 we explain why we choose the depth interval from 40 to 200 km.

*Line142 says: "In the gravity field, a gravity high with a magnitude of _40 mGal is observed over the Alps." This is not what is shown in the data. Later, I see this approaches what is on Figure 4, but there is no mention here about the forward model calculations, how they were done, how comparable are the results to the observed data. Why is this 40 mGal if the data showed a negative anomaly approaching -200 mGal (Figure 1). Is this related to the model definition between 0-70 km depths? (Or, if the goal is not to compare it to observation, but simply to give estimates of slabs' gravimetric effects, then this needs to be clearly stated – and in this case it is not clear why gravity data is presented in Figure 1?)*
The text in the manuscript might not be clear enough. In line 142 we refer to the forward calculated gravity field. However, it might be misdealing to use the word observed here and can cause confusion with the observed gravity field. We updated the manuscript to avoid confusion.

*5) LitMod models*
*There are a few elements of LitMod model definitions that would be worth better arguing, or at least describing.*
Added a sentence explaining the output of LitMod.

*Line 250 and around: this is too succinct, please discuss why these model compositions were chosen (how were those models assembled, do they refer to continental material?), and how well they represent the Alps. See also around Line280-282 (e.g., why is a Proterozoic slab composition selected?).*
We do not aim to solve the Alpine slab puzzle, we try to estimate how sub-crustal density, temperature and compositional variation related to subducting material influence the gravity response. We also try to estimate how sensitivity gravity is to those variation and finally what bias gravity model introduce when ignoring such sub-crustal variations. We reformulated large parts of the manuscript to makes this clearer (Abstract, Introduction, result and conclusion).

Line 263 and around: choosing the LAB to define models is a different way that what was presented for constant-density models. Are the model results going to be comparable?
We give a in-depth reply on the comparability of the Tesseroid and LitMod models below at the comment regarding L390-397

*In particular, it is quite surprising to read about the constant LAB depth choice as a reference. Sparse data and that it is in discussion is a weak argument not to consider those data – the situation is the same for tomographic models. And if Artemieva's model shows +/-20 km variations in depth, why not include those?*
The LAB in LitMod is defined as a thermal isoline. We need to subdivide our model space as well as the slab segment in a lithospheric and sub-lithospheric part in order to avoid that the

thermal isoline descent along the slab segment and as result the 1300°C thermal isoline would sit beneath the lower boundary of the slab segment. We discussed this in greater detail in our reply to Carla Braitenberg's comment in the general comment section above.

*Line 284 mentions a -100_K temperature anomaly. Is it for the slab? (Sentence ending on L285 says sub-litho.). Is this anomaly kept constant across the entire slab? How reasonable this is compared to the geodynamics of the Alps? Why aren't the thermal equations resolved if this is said to be an option in LitMod? Subsequently, models 7 and 8 are described to do include temperature variations (in space? I assume: : :), but Table 2 says T-anomaly of -200_K. What is the situation, then?*
We aim to quantify how large the influence of thermal and compositional parameters is due to variations in the gravity signal. Therefore, we test pure compositional variations as well as pure thermal variations. We estimate the residuals to a background model without these variations in order to obtain the variation in the gravity signal. In the updated manuscript we make this statement clearer. Also, we added subfigures to figure 12 showing now the effect of a -100 K thermal anomaly and a -200 K thermal anomaly. This was previously missing.

*Figure 8 shows vertical slabs, and with different thicknesses. Why these two choices?*
The thickness is the same for both slab segments. In figure 8 (also figure 10) a profile along the 11-degree longitude is displayed. The slab segments appear to have different thickness because the profile line is not perpendicular to the slab segments.
Regarding the vertical slab see reply to Carla Battenberg's remark in the general commend section (page 2).

*6) Discussion*
*The discussion is a mostly fair description of the results, stating some of the difficulties, limitations, and unresolved elements. Yet there are a few statements that can be contested:*
*L387 says: "For all three modelling approaches (section 3) a measurable gravity effect of the subducting slab segments is seen". I agree that all models produce gravity anomalies that are measurable. But I have doubts that this signals are distinguishable within the total field, when one considers field observations. This is even more critical as the anomaly levels vary a lot between constant-density and LitMod-based models. (Moreover, section 3 is not about modelling approaches, is that number a typo?)*
The reference to the corresponding section is corrected.
Regarding the measurable gravity signal and the distinguishable in the of those signals in the total field: that is the core statements we try to achieve in the paper. We find that ~40 mGal gravity response of subducting sub-crustal structures is significant enough to considered in density modelling of the Alps, which was before often neglected e.g. Ebbing et al., 2006; Spooner et al., 2019. For the presented study we cannot simply compare our modelling to the observed gravity field, because we neglect the contribution of the crust, topography and any heterogeneity within the crust and mantle (with the exception of approach 1 converting seismic velocities to density). We encourage integrated density and petrological modelling including subducting slab segments with our findings. Those more sophisticated models require correct crustal density and crustal thickness models as well as more sophisticated knowledge of the Alpine petrology.
We have reformulated major parts of the abstract, introduction, discussion and conclusion to make the above statement clearer in our manuscript.

*L388 says: "The independent slab segments are distinguishable to a certain degree". I think this statement is over-interpreting the results. If the images would be shown.*
*to someone not familiar with deep structure of the Alps suggested from seismology, I have doubts whether that person would point to independent slab segments. See also previous comment.*
Here we generalise all our model results too much. An important separation between approach 1 (converted densities section 3) and the tesseroid (section 4.1) and the LitMod models

(section 4.2) has been made in the updated manuscript. The gravity signal is in fact not well distinguishable in approach 1 (converted densities fig. 4). However, our statement holds up for the tesseroid and the LitMod models (Fig 6,7,11 and 12), with of cause the exception of the bivergent slab configuration in the Eastern Alps.

*L390-397: here the comparison is between constant-density and LitMod model results, but the way these models were defined (tomography contours from 70 km depth, resp. from LAB at 100 km depth) raises questions on comparability.*
In both models the slab segments are onset to the Moho interface. In LitMod we need to introduce a (technical) LAB as the LAB is defined as the 1300°C thermal isoline and we need to avoid that the isoline is following the lithospheric structure and would effectively sit beneath the slab segment. We have discussed this above e.g. in our reply to Carla Braitenberg. It is worthwhile to point out that the Tesseroid model with a constant density contrast to the surrounding material is a strong simplification of nature, as we do not consider any temperature and pressure variation with depth. The LitMod models are a more sophisticated modelling approach, which considers temperature and pressure distribution in the subsurface. We went from a simple model approach with Tesseroids to a more complexed one with LitMod. Of cause both approaches are not full consistent with each other (more parameter in the LitMod model), which always raises question of comparability. However, surprisingly enough the forward calculated gravity response of both approaches is in a similar order of magnitude.

*L401-407: here is a clearer message on why these calculations are useful. Maybe consider taking this into the intro?*
We followed this suggestion and included the mentioned paragraph in the introduction section.

*L415-417: the sharpness is pre-defined in the models, especially for temperature (which affects densities), and reality is probably less sharp than this.*
We use our modelling to estimate the influence of thermal and compositional parameters by varying those parameters within the model. The aim is not to recreate reality of the Alpine subsurface.

*7) Writing*
*Although the message of the paper can be mostly followed and understood, there is some variability in the level of information and in being "to-the-point". A proofreading after revision could smooth these out. Information pertaining to the same topic are sometimes found in rather different parts of the manuscript, these could be better grouped. An example: which method is used to calculate synthetic gravity anomalies, and what kind (how many) different model resolutions have been used. L135 says 0.2_* 3 km but does not mention the method, L197 mentioned the method but has 0.2_ * 20 km. Many statements say slabs extend to 200 km depth, L199 says 220 km. It is a bit difficult to keep up with these information bits.*
We reformulate in the updated manuscript the introduction to make our goals more to the point and introduce at an early state of the manuscript that we test 3 different approaches to characterize the gravity signal of subducting slab segments.
Regarding L135 and 197 that are two different modelling approaches, with different discretisation.
Regarding L199, we recalculated those models to have a uniform depth of 200 km to increase consistency through the different modelling approaches. In depth explanations are provided above in the reply to Carla Braitenberg's comment for L199. In addition, there is an estimation provided to how much the gravity signal changed from the original to the updated manuscript.

**Minor comments**
*-Line14-15: unclear what is "crustal thickness at 40 km depth" – please resolve this oxymoron.*
The abstract is completely reformulated

*-I find the abstract could be more specific and to-the-point. Which kind of gravity data is used and how to constrain the models? What are those significant pattern differences, and which model seems more realistic?*

It's clear from both reviews, that the abstract needed reformulation. The new abstract formulates more precise the aims of the manuscript and introduces the three different modelling approaches better.

*-L31-33: revise unclear sentence: "A major role: : : plays the Adriatic microplate"*

Reformulate corresponding sentence

*-L34: Adria's rotation as seen by GPS could be cited here.*

Added to the manuscript

*-L50: before mentioning E. Alps, maybe mention the size of the potential slab gap between C. and E. Alps?*

Included the slab gap for completeness.

*-L51: correct "is been"*

Corrected

*-L51-54: briefly explain on which observables this "classical" view is based; and, then, which were the arguments for challenging these.*

We believe the manuscript states that to a satisfactory degree. Increased geological / tomographic description of the Alpine region will not benefit the manuscript as we estimate sensitivity of gravity data to synthetic slab segments. We do not aim to recreate or solve the slab puzzle in the Alps

*-L59: dual subduction was first proposed by Prague group, in the early 1990's, and not by these two very recent papers. Several publications from the Prague group discuss this in detail, please mention their interpretation*

Representative papers by the Prague group are now cited.

*-L67: correct "dominate"*

Corrected

*-L66 and Fig. 1a: a much clearer description of what the Bouguer anomaly map is needed. What is XGM 2019? How was it obtained? What kind of resolution to expect? Or refer to what is written below. On Fig. 1b please make the fault lines more apparent (thicker line, or another colour).*

Line width increased in the figure, explanation of the key parameter for the Bouguer correction added to the caption. Changed the arrangement of the figure, topographic map is now on the left side, while the Bouguer Anomaly map is on the right.

We reformulated the Abstract as well as the Introduction to clarify the usage of the Bouguer anomaly, which is used to motivate our study rather than comparing the gravity signal of synthetic sub-crustal slab models to the observed gravity field.

*-L74-79: a better and more detailed description of the motivation, approach and goals would be very welcome here. For example, petrology is not mentioned here at all.*

As the pervious reply states, we reformulate the abstract and introduction sections to better introduce our motivation, approaches and goals.

*-L81-83: please add how this model was obtained. Satellite (which satellite?) data only? Or also land data? To make it resolved at 25 km, what assumptions were taken? Local isostatic equilibrium of the topography? (If yes, is it surprising to see isostatic equilibrium in line 88?)*

We added a more in-depth explanation to the XGM 2019 model and the way we calculate the Bouguer Anomaly map in section 2.

*-L90: use of GOCE should be mentioned earlier. What was the purpose of using these gradients?*
we anticipated that the gravity gradients at satellite altitude are sensitive to the slab segments. Our finding shows, that in fact, gravity gradients at satellite height are not sensitive to the different slab segments, and therefore we moved research corresponding to the gradients in the appendix. We believe that those findings are still significant to the manuscript and therefore, worthwhile keeping. We included this statement now in the introduction and make more clearer what our goals are within this manuscript.

*L101: correct the reference year*
Corrected

*-L105: how is the model constructed between Moho depth and 70 km depth level?*
Here we describe the MeRE2020 model which we use form 70km to 200km. Slab segments above 70 km are defined using Spada et al. 2013 crustal thickness map, with the exception of approach 1 (section 3) reasoning on the depth interval is given in the reply to Carla Braitenberg's comment L105 & L144.

*-L106: "depth-dependent average shear-wave velocity 1-D model" – is there some redundancy in the description?*
Changed to: "1-D average shear wave velocity model"

*-L108-111: the choice of 200 km depth as bottom of the model, and that anomalies below this level, seems ad hoc. Would it be possible to quantify/to justify? The El- Sharkawy et al. results extend to 300 km depth.*
Below 200 km slab segments are not well imaged anymore in the MeRE2020 model. As we want to estimate the gravity response of slab segments in the upper mantle and estimate the sensitivity of gravity to those structures, we cut our model at 200 km depth. Significant gravity contributions from slab segments below 200 – 250 km to the Alpine gravity are unlikely

-L117: see earlier comment on dual subduction and its references.
See the comments above.

*-L139: the label says Vsv is used, is Vsh also available? Why choosing one rather than the other?*
Yes indeed, only Vsv was available to me.

*-L165: Please correct "crustal mantle boundary".*
corrected

*-L192 and 198: what is the resolution of the MeRE2020 tomographic model used here, and how does it compare to 0.2_ horizontal and 20 km vertical tesseroid size?*
We interpreted the MeRE2020 tomographic model and identified the upper slab boundary at Moho depth, 100km, 150km and 200km depth. We then interpolated between the identified slab boundaries to obtain a continuous upper slab boundary. This slab model is then transferred in a tesseroid model with the discretion of 0.2° and 20km vertical resolution. We do not relate here in any from on the MeRE2020 as stated in the manuscript. The resolution of the MeRE2020 is not important to the gravity forward calculation here.

*-L197: finally comes the calculation details. I think this deserves defining a separate sub-section.*
Subsection added

*-L206: is it density anomaly instead of density?*
changed to density contrast

*-L210: uncertainty is mentioned for the first time. I agree this is important and I'm happy to see this word here. But could you please refer to uncertainty of gravity, models, tomographies also earlier in the manuscript, so that it does not come as a surprise here, and we know why parameter values were chosen as chosen?*
The numerical uncertainty of the forward gravity modelling is well beyond the uncertainty of the Alpine gravity field (see Götze et al. – Alpine Gravity Research Group). Please note again, that this manuscript is an attempt to estimate the contribution of the slabs to the gravity field, not a detailed lithospheric scale model. The uncertainties here are related to the velocity-density conversion and the definition of the slab geometries as defined in the text.

*-L211: it seems that a new section starts here, with results? If yes, please clearly state it (new sub-section). For the first model result presented here (drho=60kg/m3, Hslab=80km): why was this particular model chosen?*
Sub section added

*-L222: is it thickness rather than volume?*
Corrected

*-L245 (twice): correct spelling Perple_X.*
Corrected

*-L246: this is not what Perple_X does, this is one thing Perple_X can do. The choice of input (here 6 oxides only), the thermodynamic databases, list of solid solutions, etc. are all user chosen. Maybe the description is simply how LitMod uses Perple_X?*
Added clarification that the Perple_X description relates to the LitMod implantation.

*-L251: correct homogene to homogeneous, sub to sub-*
Corrected

*-L275: this could be presented much more nicely, with several columns representing the type of information (lithosphere, sub-lithosphere, slab config., T anomaly, etc.) and the lines showing the information itself. From the current version it is very difficult to have an overview of tested models.*
Updated table follows the above suggestions.

*-L295: is this the beginning of results? If yes, mark it with a sub-section, for example.*
Subsection added

*-L306: please clarify what causes these <1kg/m3 variations, temperature?*
temperature and pressure variation with depth. Clarified in the manuscript.

*-Figure 11: please match "Configuration" in caption and "Hypothesis" in the figure.*
Corrected in figure 11 as well as in figure 17 for the gzz component

*-L372: typo in mantel*
Corrected

*-L386: how realistic is this -100_K anomaly, and, therefore, the 16 mGal anomaly?*
We present here a scaling how much thermal and compositional variations influence the density structure and consequently the gravity response. We present how much the gravity

response changes to a thermal anomaly of -100 K and now also to -200 K to illustrate stronger the thermal effect on the density structure. Our goal is to illustrate the variance in gravity modelling, which result by a variance in thermal parameter and potentially the bias which is included into gravity models by ignoring or choosing wrong thermal parameters, rather than recreate the true thermal conditions in the Alpine region.

*-In the conclusions, I'd recommend mentioning that future results based on AlpArray Seismic Network data will be of high interest in better defining slab geometries AND properties.*
Included in the conclusion

*Appendix*
*-L634-635: some references to the data would be useful.*
Here we describe the choice of satellite height for the forward calculated gravity gradients. No dataset is included here. We reformulated this part to match better the updated abstract and introduction.
Measured gravity gradients by the GOCE satellite are discussed in the following paragraph and proper reference to the data set is already given (Bouman et al., 2016)

*-L637: correct "longwave length"*
Corrected

*-L641-645: are the obtained results in E on the same order of magnitude that you would expect? What do these maps mean, then? What is the support for the interpretation on lines 650-652 (positive signal of 0.5 E could be linked to slabs)?*
Motivation of gravity measurements are now given more clearly in the abstract, introduction as well as the beginning of the appendix. The doted lines represented the outline of the upper slab boundary. We removed those lines, as they do not add any valuable information and may course confusion, especially since I failed to mention them in the caption.

*-Figure 14: please use the same colour scale and range as on Figure 13.*
Figure 13 corresponds to the measured GOCE gradients, while Figure 14 corresponds to the forward calculated gravity field obtained by the conversion of seismic velocities to densities. The content of the gravity fields is not equivalent as we nullified any gravity contribution from the surface to 70 km, as well everything above 200 km.

---

## Author Comment (AC2) · 17 Dec 2020

Please see attached pdf. We use formatting to separate our responses from original reviewer comments. We response to both reviewer in the same pdf document.

Please also note the supplement to this comment:
https://se.copernicus.org/preprints/se-2020-145/se-2020-145-AC2-supplement.pdf
* * *

---

## Referee Report (RR1)

**Review of the Manuscript "Gravity Effect of Alpine Slab Segments Based on Geophysical and Petrological Modelling" by M. Loewe et al, submitted to Solid Earth. Review of First Revision.**

**Reviewer: Prof. Dr. Carla Braitenberg, Trieste University, Italy.**

**General Comment**

This is the second round of review of the manuscript " Gravity Effect of Alpine Slab Segments Based on Geophysical and Petrological Modelling", Lowe et al.

The aim of the study is to model the gravity field of subducting lithospheric slabs, the geometry and the density of which is formulated assuming different assumptions and hypotheses on geometry and density. The variables that have been studied include compositional aspects, temperature, and geometry of the slabs. Furthermore, a simple first order forward calculation of the gravity effect of the mantle down to a depth of 200 km is made, based on a seismic tomography model. In this case, the density model of the mantle is derived by applying single conversion factor to the percentage seismic velocity variations to obtain corresponding density anomalies. Compared to the first version, the scope and procedure of the work are now much clearer. There are still a few issues, that partly had already been mentioned in the first review, that could be improved for greater clarity. To my view there is an error on Equation 2. Detailed comments are given below.

The manuscript can be accepted pending minor revision.

**Specific corrections**

L. 23: positive gravity signal of up to 40 mGal
-> your modeling shows that the density contrast could also be negative- here it would be more correct to write ..predict a positive or negative gravity signal of up to 40 mGal....?

L. 42: Subducting lithosphere has a higher density than the surrounding mantle material at the same depth interval

In the modeling you show that density of the slab can be lower than the surrounding mantle, if very old compared to a tecton mantle. So there is the possibility that density can be lower, and is not always a positive contrast, so this sentence is misleading and does not reflect the modeling. Please adjust.

L. 50 Topography-> tomography

L. 56: Alpine gravity field have not considered any slab segments, rather they only account for the thickness of the lithosphere (e.g. Ebbing et al., 2006; Spooner et al., 2019; Tadiello and Braitenberg 2020).

-> as it stands the sentence is wrong, because in the recent work of Tadiello and Braitenberg (2021) the subcrustal seismic tomography is converted into densities and the effect is fully calculated down to the depth of the availability of the tomographic model (200 km). The seismic velocity variations have not been interpreted as slabs, but have been used to calculate the full gravity effect of the mantle. Therefore, I propose to change the sentence to the following:

Secondly, previous Alpine models addressing the Alpine gravity field have considered the subcrustal mantle inhomogeneities in form of lithosphere thickness (e.g. Ebbing et al., 2006; Spooner et al., 2019) or in form of mantle density variations (Tadiello and Braitenberg 2021), but without identifying the isolated effect of subducting slabs segments in the velocity or density variations.

L.57/58: If the contribution of the slab is not considered,

-> See above comment: the important thing to consider is the mantle density variation, if it is identified as a slab or not is a matter of interpretation. I propose to make the sentence consistent:

If the contribution of the mantle density variations are not considered, a significant part of the gravity field might be attributed to crustal thickness variations or intra-crustal sources.

L. 62: . Therefore,

L. 67: *XGM 2019*

*-> give reference*

L. 76 contrast -> contrasts

L. 105: approximately -> approximate

L. 130: We calculated the gravity contribution of the topography and bathymetry

-> give maximum calculation radius of topography for each grid point.

L. 131: Tesseroids

-> is this the software name or the object? In either case add reference

L. 134: regridded

L. 137: an isostatic compensation of the topography

-> without calculating the isostatic equilibrium you don't know if topography is compensated. More precise would be:

an isostatic crustal thickening in response to topography

L. 139: topographic correction for the gravity gradients at a station height of 225…

-> also here please give calculation radius.

L. 156: b) crustal depth estimation after Grad et al. (2009)

-> Please uniform "crustal depth" with crustal thickness used in a) – crustal depth is not the correct word. It would be Moho depth or bottom crustal depth? Geologically crustal thickness and Moho depth are not the same thing.

L. 163: descripted  -> described

L. 191: attention, eqt. (2) seems wrong.: a percentage deviation is adimensional. Please check- I suppose you mean:

rhoRel=[Vsvabs(1+delta%)-Vsvabs]*Zeta= Vsvabs* delta%  * Zeta

L. 193: *divagation* -> deviation?

L. 209: please define horizontal extension of the mantle model, and mention how you deal with border effects.

L. 229: Secondly, we create a set of slab models accounting for compositional and thermal variations with depth (approach 3). Those models are created with  the software package LitMod 3D (Fullea et al., 2009)

-> please add that in approach 3 slabs are strictly vertical due to software limitations.

"Those models" is ambiguous- please change to:

Secondly, we create a set of slab models accounting for compositional and thermal variations with depth (approach 3). The models of approach 3 are created with  the software package LitMod 3D (Fullea et al., 2009) and here the slabs are strictly vertical due to software limitations.

L. 314: Maybe you could mention that calculated field is quite different from the field of the complete mantle density inhomogeneity of Fig. 4, which only reaches a positive mantle effect of maximum 50 mGal.

L. 354: Add hear for clarity that slabs are extending vertically downwards.

L. 384: topography or crustal thickness variation are not considered

-> add for clarity: topography, crustal thickness variation and mantle variations outside the slab are not considered.

L. 390 surround- > surrounding

L. 404: Title Fig. 10a,b: profil-> profile

L. 410: contrast is  *limit to the* -> contrast is  *limited to the*

*L. 416:* significant larger-> significantly larger

L. 532: Even though this might be  considered as an end of the envelope calculations,

-> please revise sentence, not sure what you wanted to say.

L. 534: Previous studies compensated  this effect by lithosphere thickness and/or intra-crustal sources, future studies should  incorporate subducting slab structures in order to provide a meaningful representation of the  geodynamic complex Alpine area.

-> see comment above- previous works have modelled the mantle densities starting from seismic velocities and inverting the mantle densities. Please reformulate.

-> for instance: The interpretation of density variations in the mantle in terms of  subducting slab structures is a means to provide a meaningful representation of the  geodynamic complex Alpine area.

Missing references in reference list:

El-Sharkawy(2020)

Tadiello and Braitenberg 2020->Tadiello and Braitenberg 2021 (accepted in Solid Earth)

Karusova et al. 2013- > probably Karousova et al?

L. 103 Piromallo and Morello, 2003 -> probably Piromallo and Morelli, 2003

Check reference Zingerle et al., 2019- webpage? Publisher?

---

## Author Response (AR2)

Ref: se-2020-145

**Title: Gravity Effect of Alpine Slab Segments Based on Geophysical and Petrological Modelling" by Maximilian Lowe et al.**

We thank the referees once again for the constructive criticism and suggestions. All points raised by the reviewer have been addressed. We hope that the new version of the manuscript is improved and can be accepted for publication.

**Specific corrections**

L. 23: positive gravity signal of up to 40 mGal

-> your modeling shows that the density contrast could also be negative- here it would be more correct to write ..predict a positive or negative gravity signal of up to 40 mGal....?

The LitMod gravity model (approach 3) with a negative gravity signal is a result of the variation in compositional parameters which we tested. We state clearly in the manuscript that we demonstrate the influence of the compositional parameters here. We do not make the claim that a slab with a lower density in respect to the surrounding mantle is likely. We changed the sentence in question to: "Forward calculations predict a gravity signal of up to 40 mGal for the…".

L. 42: Subducting lithosphere has a higher density than the surrounding mantle material at the same depth interval In the modeling you show that density of the slab can be lower than the surrounding mantle, if very old compared to a tecton mantle. So there is the possibility that density can be lower, and is not always a positive contrast, so this sentence is misleading and does not reflect the modeling. Please adjust.

The same response to the previous comment applies here. We changed the sentence in question to: "For lithosphere to subduct, a higher density than for the surrounding mantle material at the same depth interval is required, causing a negative buoyancy for the slab…"

L. 50 Topography-> tomography

Corrected

L. 56: Alpine gravity field have not considered any slab segments, rather they only account for the thickness of the lithosphere (e.g. Ebbing et al., 2006; Spooner et al., 2019; Tadiello and Braitenberg 2020).

-> as it stands the sentence is wrong, because in the recent work of Tadiello and Braitenberg (2021) the subcrustal seismic tomography is converted into densities and the effect is fully calculated down to the depth of the availability of the tomographic model (200 km). The seismic velocity variations have not been interpreted as slabs,

but have been used to calculate the full gravity effect of the mantle. Therefore, I propose to change the sentence to the following:

Secondly, previous Alpine models addressing the Alpine gravity field have considered the subcrustal mantle inhomogeneities in form of lithosphere thickness (e.g. Ebbing et al., 2006; Spooner et al., 2019) or in form of mantle density variations (Tadiello and Braitenberg 2021), but without identifying the isolated effect of subducting slabs segments in the velocity or density variations.

Incorporated the suggested sentence.

L.57/58: If the contribution of the slab is not considered,

-> See above comment: the important thing to consider is the mantle density variation, if it is identified as a slab or not is a matter of interpretation. I propose to make the sentence consistent: If the contribution of the mantle density variations are not considered, a significant part of the gravity field might be attributed to crustal thickness variations or intra-crustal sources.

Incorporated the suggested sentence above.

L. 62: . Therefore,
corrected

L. 67: XGM 2019 -> give reference
Added reference to caption of Figure 1.

L. 76 contrast -> contrasts
corrected

L. 105: approximately -> approximate
corrected

L. 130: We calculated the gravity contribution of the topography and bathymetry
-> give maximum calculation radius of topography for each grid point.
The mass correction for the Bouguer Anomaly map was performed using Tesseroids. No specific terrain correction was carried out. We adjusted the sentence to make it clearer. The new sentence is: "The Bouguer Anomaly is calculated from the Free-Air gravity disturbance with a correction density of 2670 kg/m3 for topography, and a correction density for water of 1030 kg/m3 for the offshore areas using Tesseroids (Uieda et al., 2016). For the tesseroids, we use the topography and bathymetry from ETOPO1 resolution (Amante & Eakins, 2009), which was regridded at a regular grid with a grid space of 25 km to match the resolution of the XGM 2019 model for a maximum degree of 719"

L. 131: Tesseroids -> is this the software name or the object? In either case add reference
Its both the name of the software as well as the object. Added reference here.

L. 134: regridded

Corrected

L. 137: an isostatic compensation of the topography -> without calculating the isostatic equilibrium you don't know if topography is compensated. More precise would be: an isostatic crustal thickening in response to topography
Followed the suggested sentence above.

L. 139: topographic correction for the gravity gradients at a station height of 225…
-> also here please give calculation radius.
Response to comment regarding L 130 applies here as well. We changed topographic correction to mass correction to be consisting with our terminology.

L. 156: b) crustal depth estimation after Grad et al. (2009) -> Please uniform "crustal depth" with crustal thickness used in a) – crustal depth is not the correct word. It would be Moho depth or bottom crustal depth? Geologically crustal thickness and Moho depth are not the same thing.
Corrected. Using now uniform Moho depth in the caption of Figure 2.

L. 163: descripted -> described
Corrected

L. 191: attention, eqt. (2) seems wrong.: a percentage deviation is adimensional. Please check- I suppose you mean:
rhoRel=[Vsvabs(1+delta%)-Vsvabs]*Zeta= Vsvabs* delta% * Zeta
adjusted

L. 193: *divagation* -> deviation?
corrected

L. 209: please define horizontal extension of the mantle model, and mention how you deal with border effects.
We avoid edge effect or border effects by using relative densities. No significant edge effects are expected and therefore no horizontal extension of the model is necessary. Added following sentence to the manuscript: "No horizontal extensions of the mantle model are introduced because relative densities are used and therefore edge effects are not expected to be significant and would only affect the outer most degrees of the model. The slab segments are located central in the model far away from possible artifact due border effects."

L. 229: Secondly, we create a set of slab models accounting for compositional and thermal variations with depth (approach 3). Those models are created with the software package LitMod 3D (Fullea et al., 2009) -> please add that in approach 3 slabs are strictly vertical due to software limitations. "Those models" is ambiguous-please change to: Secondly, we create a set of slab models accounting for compositional and thermal variations with depth (approach 3). The models of approach 3 are created with the software package LitMod 3D (Fullea et al., 2009) and here the slabs are strictly vertical due to software limitations.
The sentence got reformulated following the suggestions above.

L. 314: Maybe you could mention that calculated field is quite different from the field of the complete mantle density inhomogeneity of Fig. 4, which only reaches a positive mantle effect of maximum 50 mGal.
Included a sentence following the suggestion above.

L. 354: Add hear for clarity that slabs are extending vertically downwards.
Included a sentence following the suggestion above.

L. 384: topography or crustal thickness variation are not considered -> add for clarity: topography, crustal thickness variation and mantle variations outside the slab are not considered.
Incorporated the suggestion above.

L. 390 surround- > surrounding
Corrected

L. 404: Title Fig. 10a,b: profil-> profile
corrected

L. 410: contrast is limit to the -> contrast is limited to the
Corrected

L. 416: significant larger-> significantly larger
Corrected

L. 532: Even though this might be considered as an end of the envelope calculations, -> please revise sentence, not sure what you wanted to say.
Changed sentence to:
"Even though this might be considered as a maximum gravity estimation of slabs, this value is significant, even compared to the observed Bouguer Anomaly low of -200 mGal along the Alps".

L. 534: Previous studies compensated this effect by lithosphere thickness and/or intra-crustal sources, future studies should incorporate subducting slab structures in order to provide a meaningful representation of the geodynamic complex Alpine area.

-> see comment above- previous works have modelled the mantle densities starting from seismic velocities and inverting the mantle densities. Please reformulate.
-> for instance: The interpretation of density variations in the mantle in terms of subducting slab structures is a means to provide a meaningful representation of the geodynamic complex Alpine area.
Reformulated accordingly the suggestions above.

Missing references in reference list:
El-Sharkawy(2020)
The tomographic model from El-Sharkawy was not published in a peer reviewed journal when the first draft of this manuscript was written. However, the model was published as part of the Doctoral dissertation. In the first draft the model was therefore cited by El-Sharkawy(2019). When the model got published in 2020 the citation within this manuscript was updated to El-Sharkawy et al.,(2020). During the process of updating the citation "et al.," was three times forgotten to add. It got updated now.

Tadiello and Braitenberg 2020->Tadiello and Braitenberg 2021 (accepted in Solid Earth)
Updated

Karusova et al. 2013- > probably Karousova et al?
Corrected

L. 103 Piromallo and Morello, 2003 -> probably Piromallo and Morelli, 2003
Corrected

Check reference Zingerle et al., 2019- webpage? Publisher?
Updated citation

---

## Author Response (AR3)

All minor spelling errors mentioned by the Editor were corrected.

L64: neccessary -> necessary

L65: anonaly -> anomaly

L87: compressional -> shortening

192: velocieties -> velocities

193: precentage -> percentage